# Accessory Minerals in the Chromitite Ores of Dzharlybutak Ore Group of Kempirsai Massif (Southern Urals, Kazakhstan): Clues for Ore Genesis

Dmitry E. Saveliev [1,*], Darkhan K. Makatov [2], Andrey V. Vishnevskiy [3] and Ruslan A. Gataullin [1]

1   Institute of Geology UFRC RAS, Karl Marks Str., 16/2, 450077 Ufa, Russia
2   Department of Geology and Exploration of Mineral Deposits, Abylkas Saginov Karaganda Technical University, Nursultan Nazarbaev Av., 56, Karaganda 100027, Kazakhstan
3   Sobolev Institute of Geology and Mineralogy, Siberian Branch Russian Academy of Sciences, Academika Koptyuga Av., 3, 630090 Novosibirsk, Russia
*   Correspondence: savl71@mail.ru

**Abstract:** The paper provides results of a detailed mineralogical study of some chromitite ores from two deposits in the Southern Urals of Kazakhstan: Almaz-Zhemchuzhina and Geofizicheskoe-VII. It is revealed that the main ore minerals are Cr-spinel with high Cr# (Cr/(Cr + Al) = 0.8–0.83), as well as serpentine and chlorite, replacing primary olivine. Chromium spinel grains contain mineral inclusions, which are distributed rather unevenly. The most common mineral inclusions are olivine (serpentine) and amphibole; phlogopite, pyroxenes, and base metal sulfides are rare. Olivine from inclusions in chromite is the highest in magnesium ($Fo_{97-98}$), and is anomalously high in nickel (up to 1.8 wt.% NiO). The closure of exchange reactions between olivine and chromite occurred in the temperature range of 700–850 °C and in the oxygen fugacity range of −1.04 ... +2.8 ΔFMQ, which most likely corresponds to the upper mantle settings of the fore-arc basin. A few tens of monomineral grains and polymineral intergrowths of platinum group minerals (PGMs) were found in chromite aggregates. Notably, monomineral grains are mainly represented by Ru, Os, and Ir disulfides, while in polymineral inclusions, iridium prevails (with widespread native phases, sulfides, and sulfoarsenides). PGM grains included in chromite are often associated with hydrous silicates: amphibole, and less often with phlogopite or chlorite. Discussed in the paper is the possible genesis of ores and inclusions. As a preliminary conclusion, we suggest that the solid-phase processes played the most significant role in the crystallization of Cr-spinel in the investigated chromitite ores.

**Keywords:** chromitite ores; Cr-spinel; olivine; amphibole; PGM; ophiolite; ultramafic rock; Kempirsai

## 1. Introduction

Kempirsai is one of the largest massifs in the Urals that hosts unique deposits of chromium ores. These deposits are the largest known ones, contained in ophiolite-type complexes, and are the second-most abundant reserves, following the Bushveld chromite deposits. Ophiolite deposits are often referred to as so-called podiform deposits due to their irregular morphology, pod-like shape and dunite framing. These features place them at a sharp contrast to reef layers, which have a persistent strike in platform-layered intrusions.

A distinctive feature of most podiform deposits is their significant manifestation of secondary processes that affect host rocks, mainly dunites, harzburgites, and lherzolites, often completely transforming them into serpentinites and rare chloritites. Deposits of the Main Ore Field of the Kempirsai massif are not an exception. Here, chromitites are usually framed by dunitic serpentinites, which are then gradually changed by peridotites with abundant bastite pseudomorphs after pyroxenes. Relatively fresh blocks of lherzolites and harzburgites occur sporadically only in some boreholes, at depths from 300 to 1000 m, and even deeper. However, the serpentinization of ultramafic rocks is usually

constrained by the low-temperature stage of mesh-textured serpentine formation, while accessory and ore-forming Cr-spinels are nearly always well-preserved and have only minor secondary changes.

The genesis of chromitites of the ophiolite association is still the subject of debate. Possible mechanisms of chromitite formation include crystal differentiation [1–3], etc.), liquid immiscibility [4], magma mixing/mingling [5,6], mantle/melt interaction [7–10], and solid state redistribution of mineral phases in a rising mantle flow [11]. Along with the development of a dominant melt-rock interaction model [8–10,12], the formation of dunite-hosted chromitite was considered to be a result of the interaction of peridotites with percolating boninitic or basaltic melts [13,14]. This idea was based on their improbable crystallization due to simple differentiation [15].

Studies of mineral inclusions in both accessory and ore Cr-spinel from ophiolites [16–20] showed that the composition only of few inclusions corresponds to those of minerals from host rocks [21]. In most cases, the composition of inclusions significantly differs from the compositions of possible parental or mantle-interacted melts [20]. These inclusions are called exotic [20], and are considered a result of multiple reactions involving mantle ultramafic rocks and percolating melts of various compositions [10], or fluids [19,20]. Chromite grains often contain solid inclusions, which are highly diverse in their composition. Along with inclusions typical of ultramafic rocks (olivine, pyroxenes, serpentine, chlorite, platinum group minerals, awaruite), there are quite a few that are non-typical for ultramafic rocks (amphiboles, phlogopite) and "exotic" inclusions (zircon, monazite, diamond, moissanite, corundum, rutile, titanite, etc.) [19,20,22,23]. Currently, the presence of such inclusions has led researchers to conclude that fluid-hydrothermal processes might take place in the genesis of mineralization all the more often.

In addition, in recent years, the joint findings of the ultra-high pressure, highly reduced and crustal minerals in chromitites have allowed a number of researchers to suggest that the formation of ores is a multistage process, including the formation of chromite at high pressures at different levels of the mantle, the introduction of crustal minerals from subducted slab, the extraction of the mineral that is associated with the deposition of chromitites in dunite conduits [22–25], etc. In our previous works, mainly on the basis of geological and structural data, a solid-state plastic flow mechanism for the formation of podiform chromitites is supported [11,26,27].

Mineralogical features of the chromitites of the Kempirsai massif have been highlighted in some previous works [28–36] with the limelight given to descriptions of PGM inclusions. The purpose of this work was to characterize mineral inclusions in grains of the ore-forming Cr-spinels from one of the most productive ore clusters in the southern Kempirsai massif, i.e., Dzharlybutak. One important goal was the high-resolution imaging of PGM aggregates, which has been difficult to achieve until recently. The inclusions are characterized based on the example of the core material from wells drilled at the Almaz-Zhemchuzhina deposit, which is unique in size, and the minor Geofizicheskoe-VII deposit. In previous works [30,32,35,36] the results of studying chromitites from the upper horizons of the Almaz-Zhemchuzhina deposit were presented; in this paper, samples from its deep levels were considered. The samples of cores on the Geofizicheskoe-VII deposit were obtained very recently and have not been studied by anyone before. Features of localization, morphology and composition of the inclusions of various minerals in chromite grains can facilitate interpretation of the ore formation processes.

## 2. Geological Background

The Kempirsai ultramafic massif is one of the largest in the Urals. It contains major accumulations of chromium ores, confined to rocks of the ophiolite association. The largest podiform chromite deposits are located in the southeastern part of the massif within the so-called Main Ore Field, where they yield two ore zones (western and eastern) and several ore clusters (Figure 1).

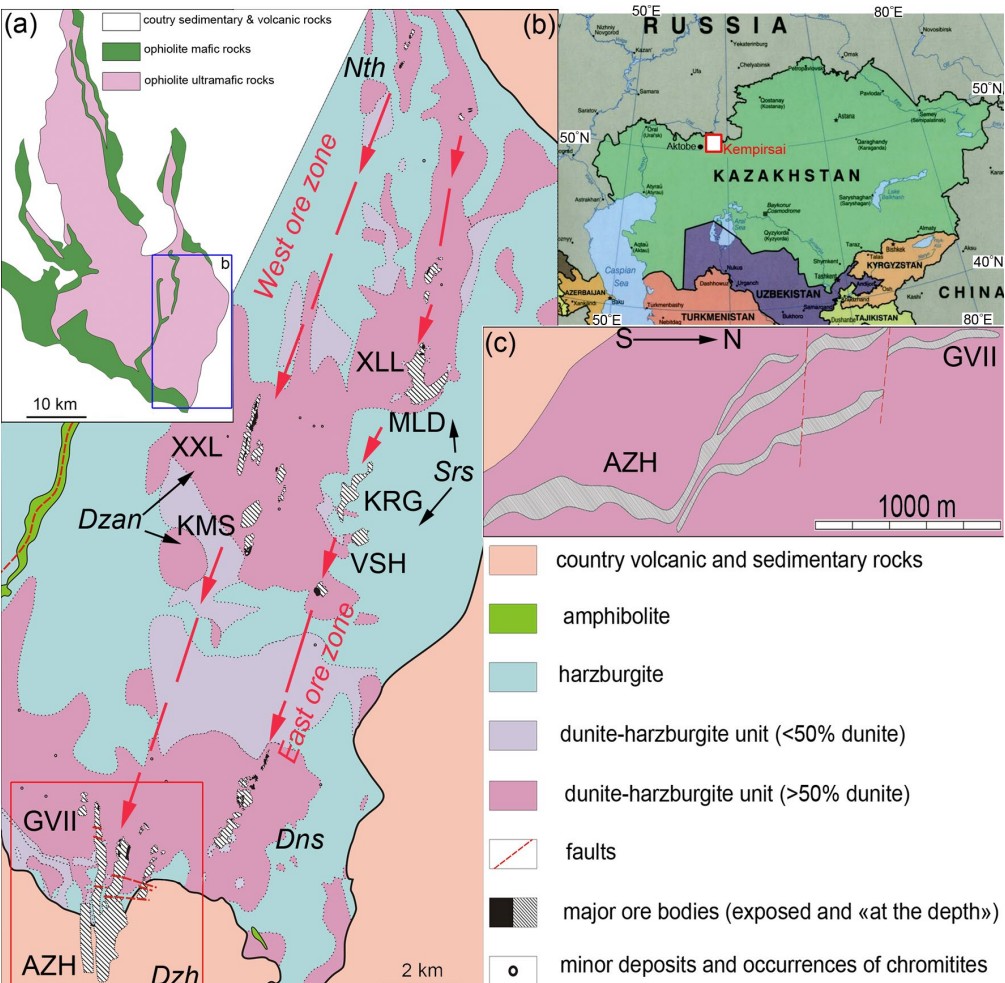

**Figure 1.** (**a**) Schematic geological map of the Kempirsai massif, (**b**) geological map of the Main Ore Field and (**c**) section through Almaz-Zhemchuzhina-Geofizicheskoe-VII deposits (after [4,37,38]). Ore clusters (italic): Dns—Donskoi, Dzan—Dzhangizagach, Dzh—Dzharlybutak, Nth—Northern, Srs—Sarysai; Deposits: AZH—Almaz-Zhemchuzhina, GVII—Geofizicheskoe-VII, KMS—Komsomol'skoe, KRG—Karaagach, MLD—Molodyozhnoe, VSH—Voskhod, XLL—XL let Kazakhskoj SSR, XXL—XX let Kazakhskoj SSR.

The Almaz-Zhemchuzhina deposit is the world's largest podiform deposit of chromitites, with ore reserves of more than 100 million tons at a depth of 1200 m below the surface [4]. It occurs as the thickest (central) branch of the Dzharlybutak ore cluster, which consists of three deposits. The other two branches are the Millionnoye (western) and Pervomayskoye (eastern) deposits. In the upper part of the ore system, branching bodies of densely disseminated and massive chromitites, with a submeridional strike and steep western dip, dominate. At the depth of 50–100 m, these bodies merge into a thicker, compact deposit, with an abundance of massive ores.

The central branch is characterized by a steeper southern dipping. In the upper part of the deposit, bodies of complex morphology with varying amounts of densely disseminated and massive chromitites dominate. At deeper levels, the deposits become simpler in shape. They usually occur as gently dipping columnar bodies, turning into a thick, wide and an almost horizontal chromitite lode [37,38]. To the south, the branch slightly narrows, while its thickness increases noticeably (up to 200 m). Massive and densely disseminated coarse-grained chromitites dominate in the structure of the deposit. The ores are often intersected by thin veins of calcite, amphibole and clinopyroxene.

The Geofizicheskoe-VII deposit is a small body in the northern part of the Dzharlybu­tak ore cluster. It occurs close to the surface and has recently been mined and explored in detail. In the structure of the deposit, massive and densely disseminated chromitites dominate; the host rocks are completely serpentinized dunites, and bodies of magnesite are occasionally observed (Figure 2).

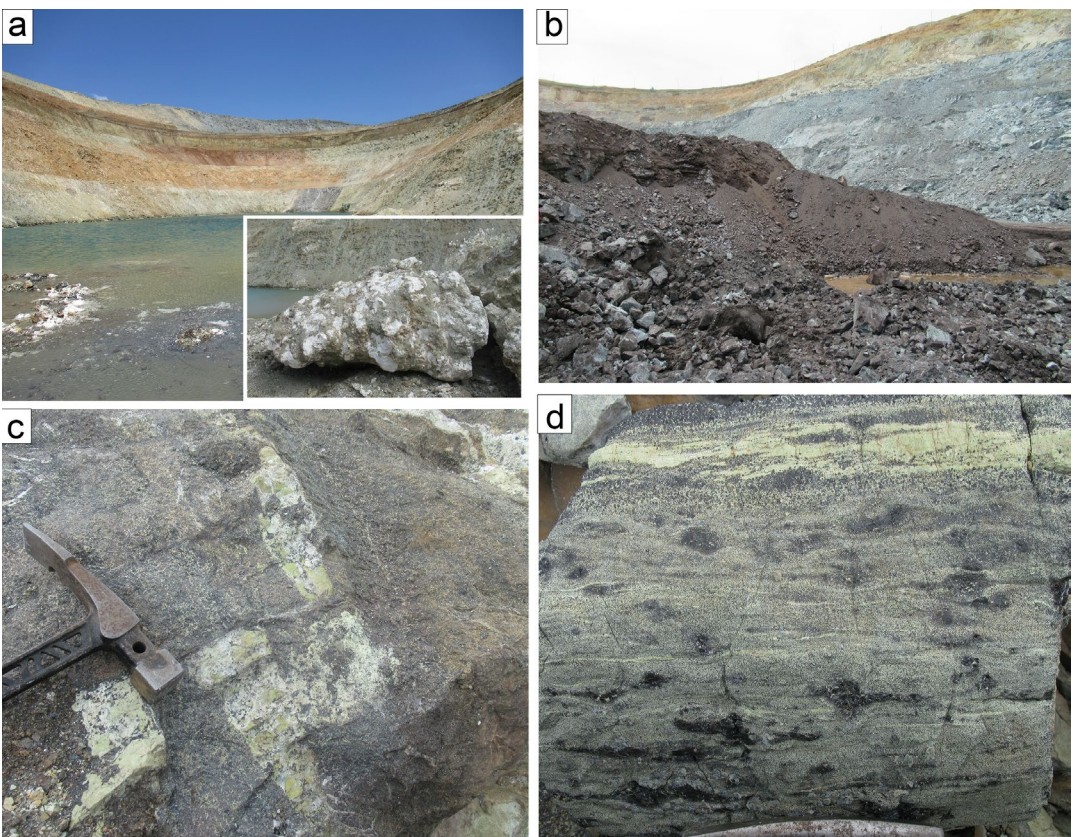

**Figure 2.** (**a**) General view of Geofizicheskoe-VII quarry and a large fragment of magnesite (in the inset), (**b**) massive chromitite deposit in the bottom of the quarry, (**c**) massive chromitite with dunite (dunitic serpentine) inclusions, (**d**) lensely-banded chromitite (Almaz-Zhemchuzhina dump).

## 3. Methods

The study objects were chromitites from two deposits of the Dzharlybutak ore cluster in the Main Ore Field of the Kempirsai massif, i.e., Almaz-Zhemchuzhina and Geofizicheskoe-VII (Figure 1). We studied polished sections of core samples from the Almaz-Zhemchuzhina (750–1100 m deep) and Geofizicheskoe-VII deposits (70–220 m deep), as well as from dumps of the Obyedinenny quarry. A total of 50 polished sections with an area of 30×20 mm were studied. Electron microscopic and compositional studies of minerals were carried out for polished sections and checkers on a Tescan Vega 4 Compact scanning electron microscope with an Xplorer 15 Oxford Instruments energy-dispersive analyzer (IG UFRC RAS, Ufa). The spectra were processed automatically using the AzTec One software package with the TrueQ technique. The elemental compositions were quantified using standard samples of natural and synthetic compounds. The following settings were used in the survey: an accelerating voltage of 20 kV, probe current in the range of 3–4 nA, spectrum accumulation time of 60 s in the "Point&ID" mode.

Structural formulae of olivine and minerals of the spinel group were calculated for 4 oxygen atoms, and those of pyroxenes were calculated for 6 oxygen atoms. For amphi­boles, we made calculations for 23 oxygen atoms by the method described in [39]. In the formulae of the spinel group minerals, the amount of bi- and trivalent iron was determined by stoichiometry. The compositions of olivine and pyroxenes were considered to deter-

mine the contents of the following minals: forsterite (Fo = Mg/(Mg + Fe), at.%), enstatite (En = Mg/(Mg + Fe + Ca), at.%), ferrosilite (Fs = Fe/(Mg + Fe + Ca), at.%), and wollastonite (Wo = Ca/(Mg + Fe + Ca), at.%). Abbreviations of minerals that we used in figures comply with those recommended in [40]. Alternatively, the minerals are designated by formulae approximately corresponding to their chemical compositions.

## 4. Results

The studied samples are represented by massive (>90 vol.% of chromite) and densely disseminated (70–90 vol.% of chromite) ores, and only two samples are rare (20–50 vol.%) and moderately disseminated (50–70 vol.%) varieties. The textures and structures of chromitites at both studied deposits have similar features: medium-grained (0.5–2 mm) ores dominate, coarse-grained (>2 mm) and fine-grained (<0.5 mm) ores are less common. The textures of the ores are mainly massive and uniform; banding was observed in samples of rare-disseminated chromitites only. Mineral inclusions were found in chromite grains in all of the studied samples (Table 1). Notably, their content rises in massive ores compared to disseminated fine-grained ones.

**Table 1.** Frequency of different minerals in the chromitites of the Almaz-Zhemchuzhina and the Geofizicheskoe-VII deposits.

| ## | Mineral | Formula | AZH | GVII |
|----|---------|---------|-----|------|
| 1 | Cr-spinel | $(Mg,Fe)(Cr,Al,Fe)_2O_4$ | +++++ | +++++ |
| 2 | serpentine | $Mg_3(Si_2O_5)(OH)_4$ | ++++ | ++++ |
| 3 | Mg-chlorite | $Mg_5Al(AlSi_3O_{10})(OH)_8$ | +++ | +++ |
| 4 | Na-Ca-amphibole | $NaCa_2(Mg,Fe,Cr)_5(Si_7Al)O_{22}(OH)_2$ | +++ | +++ |
| 5 | orthopyroxene (enstatite) | $MgSiO_3$ | +++ | − |
| 6 | olivine (forsterite) | $(Mg,Fe)_2SiO_4$ | ++ | ++ |
| 7 | clinopyroxene (diopside) | $CaMgSi_2O_6$ | + | − |
| 8 | garnet (uvarovite) | $Ca_3Cr_2(SiO_4)_3$ | − | + |
| 9 | phlogopite | $KMg_3(AlSi_3O_{10})(OH)_2$ | + | − |
| 10 | zircon | $Zr(SiO_4)$ | + | − |
| 11 | monazite | $(La,Ce,Nd)(PO_4)$ | + | − |
| 12 | apatite | $Ca_5(PO_4)_3(F,OH)$ | + | − |
| 13 | kassite | $CaTi_2O_4(OH)_2$ | + | − |
| 14 | pentlandite | $(Fe,Ni)_9S_8$ | − | ++ |
| 15 | Co-bearing pentlandite | $(Fe,Ni,Co)_9S_8$ | − | + |
| 16 | heazlewoodite | $Ni_3S_2$ | ++ | ++ |
| 17 | millerite | $NiS$ | ++ | − |
| 18 | chalcocite | $Cu_2S$ | ++ | − |
| 19 | native copper | $Cu$ | + | + |
| 20 | native nickel | $Ni$ | + | − |
| 21 | awaruite | $Ni_3Fe$ | ++ | ++ |
| 22 | laurite | $(Ru,Os,Ir)S_2$ | ++ | ++ |
| 23 | erlichmanite | $(Os,Ru,Ir)S_2$ | ++ | ++ |
| 24 | cuproiridsite | $CuIr_2S_4$ | + | + |
| 25 | irarsite | $(Ir,Ru,Rh,Os)AsS$ | + | + |

**Table 1.** *Cont.*

| ## | Mineral | Formula | AZH | GVII |
|---|---|---|---|---|
| 26 | osarsite | (Os,Ru,Ir)AsS | + | + |
| 27 | ruarsite | (Ru,Os,Ir)AsS | − | + |
| 28 | Ni-Cu-Ir-S phase | (Ir,Ni,Cu,Fe)S | + | + |
| 29 | kashinite | $(Ir,Rh,Ni,Cu,Fe)_2S_3$ | + | + |
| 30 | native ruthenium | (Ru,Os,Ir,Fe) | − | + |
| 31 | native iridium | Ir | + | + |
| 32 | Os-iridium | (Ir,Os) | + | + |
| 33 | Ir-osmium | (Os,Ir) | + | + |

+++++—main mineral (>50%), ++++—subordinate mineral (10%–50%), +++—accessory mineral, ++—rare mineral, +—few incidences of the mineral, −−—not detected; AZH—Almaz-Zhemchuzhina, GVII—Geofizicheskoe-VII.

The main ore mineral that was observed was Cr-spinel, where the content of $Cr_2O_3$ varies from 57 to 65 wt.%, $Al_2O_3$ contents range from 8.2 to 10.56 wt.%, $FeO + Fe_2O_3$ contents range from 13.5 to 19.31 wt.%, and MgO contents range from 13.5 to 14.75 wt.% (Table 2). As for minor elements, significant amounts were found only for $TiO_2$ (0.16–0.46 wt.%) and NiO (up to 0.24 wt.%); the concentrations of the rest of the minerals were below the detection limit. The chromite grains are compositionally homogeneous and always showed a very high #Cr (0.79–0.83) compared to the accessory Cr-spinels of the wall rocks (Figure 3a,b). Massive chromitites produce aggregates of closely intergrown grains, where boundaries of individual crystals are almost indistinguishable. Interstices of the ore aggregates are usually filled with secondary silicate minerals, i.e., chlorite or serpentine.

**Table 2.** Compositions of ore-forming Cr-spinels of Almaz-Zhemchuzhina and Geofizicheskoe-VII deposits.

| wt.% | Almaz-Zhemchuzhina | | | | | | | | | | Geofizicheskoe-VII | | | | | |
|---|---|---|---|---|---|---|---|---|---|---|---|---|---|---|---|---|
| MgO | 13.39 | 13.28 | 14.92 | 13.72 | 14.33 | 14.42 | 14.09 | 14.00 | 13.77 | 13.87 | 13.65 | 13.53 | 15.12 | 14.56 | 14.15 | 14.15 |
| $Al_2O_3$ | 9.66 | 10.32 | 8.88 | 8.79 | 9.04 | 8.64 | 8.85 | 10.56 | 9.56 | 9.43 | 9.41 | 9.59 | 9.29 | 9.16 | 8.86 | 9.03 |
| $TiO_2$ | 0.22 | 0.46 | 0.20 | 0.18 | 0.17 | bdl | bdl | 0.28 | 0.23 | bdl | 0.19 | 0.16 | bdl | 0.17 | 0.18 | bdl |
| $Cr_2O_3$ | 62.16 | 61.97 | 63.18 | 62.64 | 62.56 | 63.13 | 63.64 | 61.03 | 62.47 | 62.07 | 60.80 | 61.29 | 63.01 | 64.22 | 64.19 | 61.14 |
| FeO | 15.23 | 13.34 | 14.58 | 14.67 | 14.79 | 14.56 | 15.62 | 14.31 | 13.74 | 15.04 | 15.80 | 15.26 | 13.49 | 13.94 | 13.28 | 13.99 |
| NiO | bdl | bdl | bdl | bdl | 0.18 | 0.24 | bdl | 0.20 | bdl | bdl | bdl | 0.17 | bdl | bdl | bdl | bdl |
| Total | 100.7 | 99.5 | 101.8 | 100.0 | 101.1 | 101.0 | 102.2 | 100.4 | 99.8 | 100.4 | 99.8 | 100.0 | 100.9 | 102.0 | 100.7 | 98.3 |
| apfu | | | | | | | | | | | | | | | | |
| Al | 0.367 | 0.395 | 0.332 | 0.336 | 0.341 | 0.326 | 0.331 | 0.398 | 0.364 | 0.358 | 0.359 | 0.366 | 0.348 | 0.341 | 0.335 | 0.349 |
| Cr | 1.582 | 1.590 | 1.582 | 1.606 | 1.580 | 1.599 | 1.597 | 1.545 | 1.597 | 1.580 | 1.557 | 1.567 | 1.583 | 1.605 | 1.630 | 1.584 |
| Mg | 0.642 | 0.642 | 0.704 | 0.663 | 0.682 | 0.688 | 0.666 | 0.668 | 0.663 | 0.665 | 0.659 | 0.652 | 0.716 | 0.686 | 0.677 | 0.691 |
| $Fe^{3+}$ | 0.052 | 0.010 | 0.081 | 0.065 | 0.064 | 0.083 | 0.080 | 0.054 | 0.019 | 0.072 | 0.069 | 0.055 | 0.057 | 0.040 | 0.029 | 0.054 |
| $Fe^{2+}$ | 0.352 | 0.351 | 0.297 | 0.325 | 0.324 | 0.298 | 0.325 | 0.322 | 0.351 | 0.324 | 0.352 | 0.352 | 0.295 | 0.324 | 0.324 | 0.323 |
| Ti | 0.005 | 0.011 | 0.005 | 0.004 | 0.004 | | | 0.007 | 0.006 | | 0.005 | 0.004 | | 0.004 | 0.004 | |
| Ni | | | | | 0.005 | 0.006 | | 0.005 | | | | 0.004 | | | | |
| #Cr | 0.81 | 0.80 | 0.83 | 0.83 | 0.82 | 0.83 | 0.83 | 0.79 | 0.81 | 0.82 | 0.81 | 0.81 | 0.82 | 0.82 | 0.83 | 0.82 |
| #Mg | 0.65 | 0.65 | 0.70 | 0.67 | 0.68 | 0.70 | 0.67 | 0.67 | 0.65 | 0.67 | 0.65 | 0.65 | 0.71 | 0.68 | 0.68 | 0.68 |

SEM EDS data. Bdl—below detection limit.

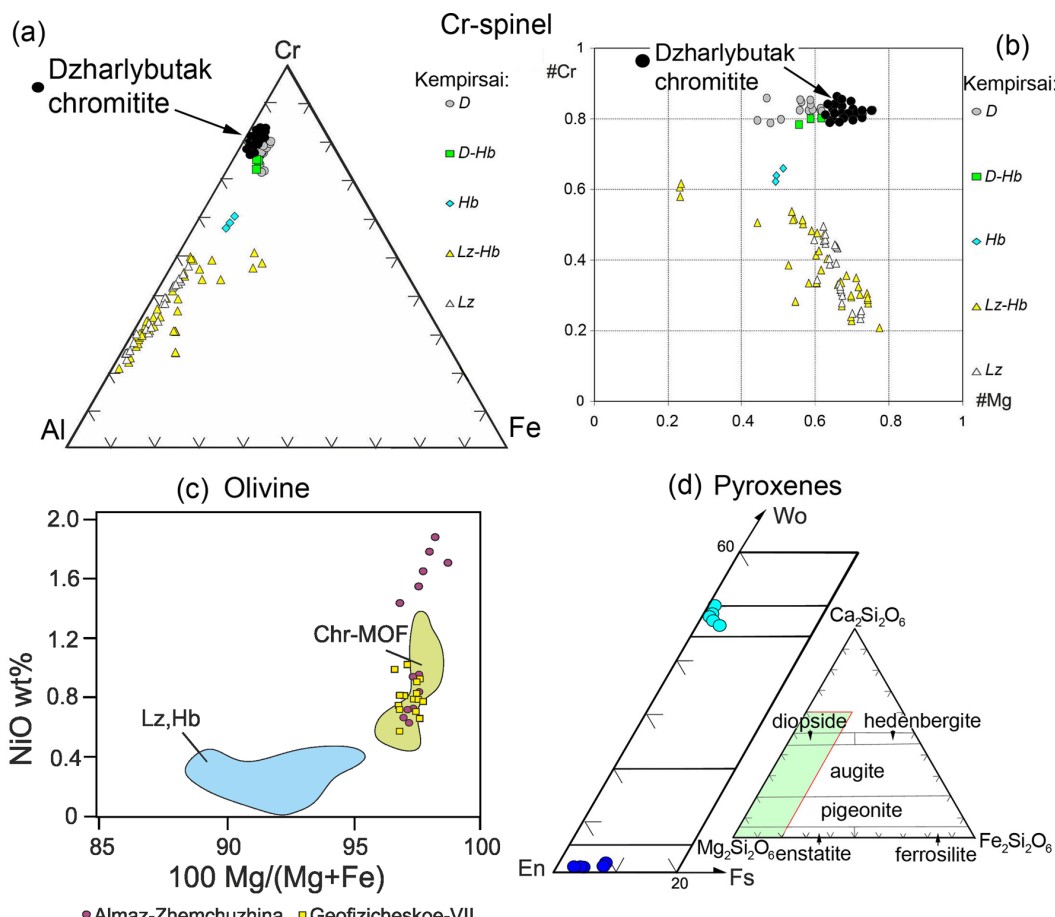

**Figure 3.** Composition of Cr-spinel, olivine and pyroxenes from Dzharlybutak chromitite. (**a**) Al–Cr–Fe diagram for three-valent cations of Cr-spinel; (**b**) #Cr = Cr/(Cr + Al) vs. #Mg = Mg/(Mg + Fe) diagram for Cr-spinel grains; (**c**) 100 Mg/(Mg + Fe) vs. NiO for olivine from inclusions in chromite grains; (**d**) the composition of pyroxenes from inclusions in chromite grains, following [41]; field in (**c**), following [32]; Lz—lherzolite; Hb—harzburgite, D—dunite, Chr-MOF—chromitite from the Main Ore Field of Kempirsai.

Mineral inclusions in the densely disseminated and massive ores are distributed unevenly. Thus, the vast majority of grains (80%) contain either no inclusions, or small ones only a few micrometers in size. Approximately 10–15% of the grains contain fairly large, single inclusions of olivine, which is currently completely replaced by serpentine. Only 2–5% of the grains show abundant inclusions of other phases, where amphibole is most common, and phlogopite is rare (size from a few microns to 50 μm) and occurs in samples from the Almaz-Zhemchuzhina deposit only. As for the inclusions of olivine and amphibole, their distribution, morphology, and preservation are similar in samples from both deposits. Other inclusions (Ni sulfides, awaruite, PGM) are rare and their sizes vary from a fraction of a micrometer to 10–25 μm. A description of these inclusions is given in the next paragraph.

Olivine in Cr-spinel grains is mainly preserved as minor (10–50 μm) rounded and oval inclusions distant from cracks (Figure 4a–c). Usually, they are intensively developed in chromitites and divide ore grains into separate fragments. However, most of the primary olivine inclusions that were observed were not preserved, because they were serpentinized during the formation of cracks in the host mineral. Compositionally, relict olivine is high in magnesium ($Fo_{95–98}$), and contains significant amounts of nickel (0.6–1.8 wt.%) (Table 3, Figure 3c). It should be noted that in the interval of 760–800 m of the Almaz-Zhemchuzhina deposit and in the interval of 120–140 m of the Geofizicheskoe-VII deposit,

olivine with anomalous NiO contents (1–1.8 wt.%) is widespread. As the depth increases, the NiO concentration falls to 0.6–0.9 wt.%. A similar phenomenon is observed at the Geofizicheskoe-VII field below and above the indicated interval.

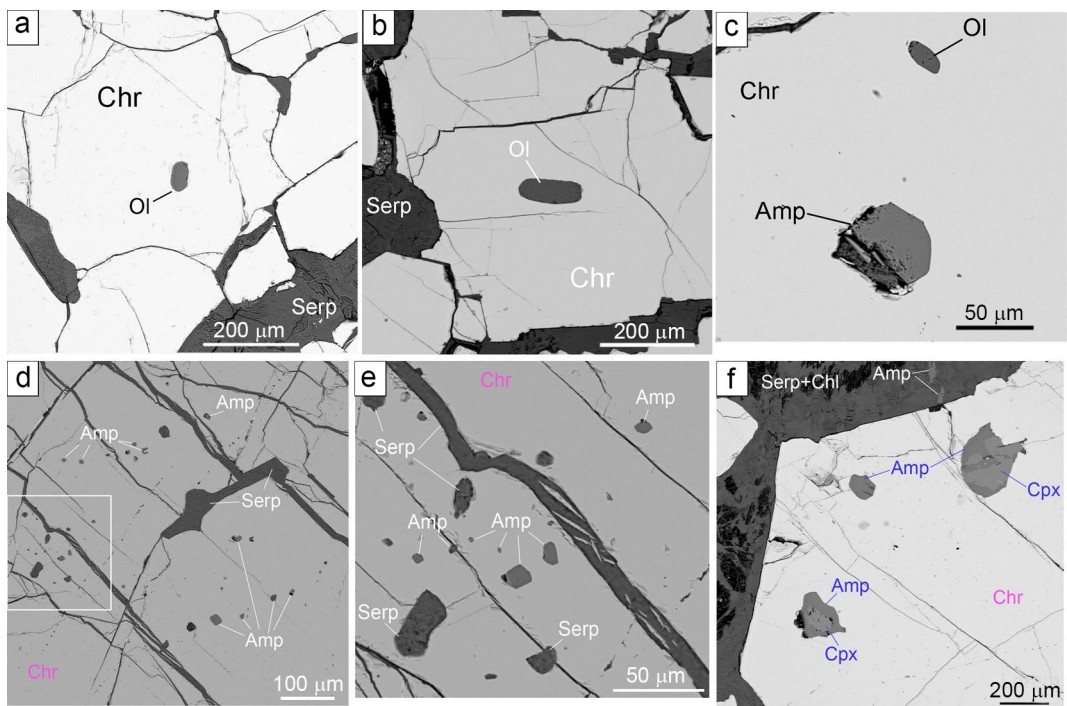

**Figure 4.** (**a**) Silicate inclusions in the chromite from deposits of the Dzharlybutak group. (**a**,**b**) Relic olivine inclusions, (**c**) neighbor inclusions of olivine and amphibole grains, (**d**,**e**) numerous inclusions of amphibole grains and olivine grains replaced by serpentine, (**d**) general view, (**e**) detailed image, (**f**) inclusions of amphibole and amphibole–diopside intergrowths. Amp—amphibole, Cpx—clinopyroxene, Chl—chlorite, Chr—chromite, Ol—olivine, Serp—serpentine.

**Table 3.** Composition of olivine from the inclusions in chromite grains of Almaz-Zhemchuzhina and Geofizicheskoe-VII deposits.

| wt.% | | | | Almaz-Zhemchuzhina | | | | | | | | | Geofizicheskoe-VII | | | |
|---|---|---|---|---|---|---|---|---|---|---|---|---|---|---|---|---|
| SiO$_2$ | 41.30 | 41.31 | 41.38 | 41.11 | 40.61 | 41.50 | 41.10 | 42.56 | 42.75 | 41.69 | 41.04 | 41.71 | 40.91 | 43.01 | 40.48 | 42.60 |
| FeO | 2.77 | 2.94 | 1.87 | 2.12 | 2.45 | 2.58 | 2.62 | 2.97 | 2.51 | 2.98 | 3.08 | 2.83 | 2.56 | 3.03 | 2.44 | 2.65 |
| MgO | 54.71 | 55.38 | 55.06 | 54.47 | 53.84 | 56.05 | 54.78 | 54.94 | 55.26 | 55.34 | 54.34 | 55.28 | 54.98 | 56.25 | 55.02 | 56.15 |
| NiO | 0.65 | 0.69 | 1.66 | 1.83 | 1.51 | 0.95 | 0.94 | 1.41 | 0.67 | 0.81 | 0.98 | 1.01 | 0.90 | 0.59 | 0.78 | 0.83 |
| Total | 99.4 | 100.3 | 100.0 | 99.5 | 98.4 | 101.1 | 99.4 | 101.9 | 101.2 | 100.8 | 99.4 | 100.8 | 99.4 | 102.9 | 98.7 | 102.2 |
| apfu | | | | | | | | | | | | | | | | |
| Si | 0.982 | 0.973 | 0.978 | 0.978 | 0.978 | 0.969 | 0.977 | 0.993 | 1.000 | 0.978 | 0.978 | 0.979 | 0.972 | 0.990 | 0.966 | 0.985 |
| Fe | 0.055 | 0.058 | 0.037 | 0.042 | 0.049 | 0.050 | 0.052 | 0.058 | 0.049 | 0.058 | 0.061 | 0.055 | 0.051 | 0.058 | 0.049 | 0.051 |
| Mg | 1.951 | 1.956 | 1.953 | 1.944 | 1.944 | 1.963 | 1.953 | 1.923 | 1.939 | 1.948 | 1.942 | 1.946 | 1.960 | 1.941 | 1.970 | 1.948 |
| Ni | 0.012 | 0.013 | 0.032 | 0.035 | 0.029 | 0.018 | 0.018 | 0.027 | 0.013 | 0.015 | 0.019 | 0.019 | 0.017 | 0.011 | 0.015 | 0.015 |
| Fo | 0.973 | 0.971 | 0.981 | 0.979 | 0.975 | 0.975 | 0.974 | 0.971 | 0.975 | 0.971 | 0.969 | 0.972 | 0.975 | 0.971 | 0.976 | 0.974 |

SEM EDS data.

The amount of pyroxenes in the investigated chromitite samples is low (Table 4, Figure 3d). Orthopyroxene (enstatite) was observed as single inclusions inside chromite grains, but it is more widespread in vein minerals, inside veinlets intersecting chromi-

tites, where it is often replaced by amphibole. Clinopyroxene (diopside) occurs as rare, small, prismatic inclusions in chromite (10–25 µm), often in association with amphiboles (Figure 4f). Both pyroxenes included in the chromite of the Almaz-Zhemchuzhina deposit are characterized by very low aluminum contents (0.29–0.87 wt.% $Al_2O_3$), while these values are significantly higher (1.85–3.04 wt.% $Al_2O_3$) for the inclusions from the Geofizicheskoe-VII chromite. The content of titanium and manganese in all studied grains is below the detection limit, and sodium is present only in clinopyroxenes from the Almaz-Zhemchuzhina deposit.

**Table 4.** Composition of pyroxenes from inclusions in chromite grains of the Almaz-Zhemchuzhina and Geofizicheskoe-VII deposits.

| wt.% | Almaz-Zhemchuzhina | | | | | | Geofizicheskoe-VII | | | | | | |
|---|---|---|---|---|---|---|---|---|---|---|---|---|---|
| $SiO_2$ | 53.52 | 54.05 | 54.78 | 57.90 | 59.12 | 59.06 | 52.31 | 53.04 | 52.97 | 55.20 | 55.71 | 55.74 | 55.29 |
| $Al_2O_3$ | 0.76 | 0.87 | 0.36 | 0.38 | 0.34 | 0.43 | 2.81 | 1.42 | 2.13 | 2.32 | 2.23 | 2.59 | 3.04 |
| FeO | 1.02 | 1.22 | 0.46 | 2.91 | 2.66 | 3.07 | 1.85 | 1.88 | 2.16 | 5.32 | 5.48 | 5.59 | 5.53 |
| MgO | 17.92 | 18.44 | 18.32 | 37.60 | 38.37 | 37.95 | 17.55 | 18.05 | 17.90 | 34.58 | 35.25 | 34.87 | 35.00 |
| CaO | 23.96 | 24.05 | 26.17 | 0.24 | 0.11 | 0.31 | 22.78 | 23.41 | 23.17 | 0.90 | 0.39 | 0.32 | 0.38 |
| $Na_2O$ | 0.41 | 0.27 | bdl | bdl | bdl | bdl | bdl | bdl | bdl | bdl | bdl | bdl | bdl |
| $Cr_2O_3$ | 1.50 | 1.09 | 0.32 | 0.48 | 0.39 | 0.28 | 0.89 | 0.42 | 0.95 | 0.51 | 0.44 | 0.60 | 0.66 |
| NiO | bdl | bdl | bdl | 0.20 | 0.31 | 0.20 | bdl | bdl | bdl | bdl | bdl | bdl | bdl |
| Total | 99.09 | 100.00 | 100.41 | 99.71 | 101.30 | 101.30 | 98.19 | 98.21 | 99.29 | 98.84 | 99.50 | 99.70 | 99.89 |
| apfu | | | | | | | | | | | | | |
| Si | 1.950 | 1.949 | 1.968 | 1.963 | 1.971 | 1.972 | 1.927 | 1.951 | 1.931 | 1.910 | 1.912 | 1.913 | 1.892 |
| Al | 0.033 | 0.037 | 0.015 | 0.015 | 0.013 | 0.017 | 0.122 | 0.061 | 0.092 | 0.095 | 0.090 | 0.104 | 0.122 |
| Fe | 0.031 | 0.037 | 0.014 | 0.082 | 0.074 | 0.086 | 0.057 | 0.058 | 0.066 | 0.153 | 0.157 | 0.160 | 0.158 |
| Mg | 0.979 | 0.998 | 0.987 | 1.912 | 1.919 | 1.901 | 0.970 | 0.996 | 0.979 | 1.795 | 1.815 | 1.795 | 1.796 |
| Ca | 0.935 | 0.929 | 1.007 | 0.009 | 0.004 | 0.011 | 0.899 | 0.922 | 0.905 | 0.033 | 0.014 | 0.012 | 0.014 |
| Na | 0.029 | 0.019 | | | | | | | | | | | |
| Cr | 0.043 | 0.031 | 0.009 | 0.013 | 0.010 | 0.007 | 0.026 | 0.012 | 0.027 | 0.014 | 0.012 | 0.016 | 0.018 |
| Ni | | | | 0.006 | 0.008 | 0.005 | | | | | | | |

SEM EDS data. Bdl—below detection limit.

Amphiboles in the chromitites were observed in two modes: (1) in the form of primary inclusions in chromite grains and (2) in interstices in association with secondary minerals, i.e., serpentine and chlorite; the second type was found only in samples from the Almaz-Zhemchuzhina deposit. The composition of interstitial amphibole corresponds to chromio-tremolite, while primary amphibole inclusions are always represented by, essentially, Na–Ca varieties: chromio-edenite, and rarely magnesian-hornblende and pargasite (Table 5, Figure 5). Notably, amphibole inclusions in chromite grains are the most numerous, with their size ranging from a few microns to 25–30 µm, and their shape varying from tabular to prismatic and acicular (Figure 4c–f). Compositionally, the inclusions are fairly consistent and high in magnesium (19.42–23.43 wt.% MgO), calcium (11.24–13.38 wt.% CaO), and chromium (1.56–3.17 wt.% $Cr_2O_3$). The content of aluminum in the vast majority of analyses (3.58–9.06 wt.% $Al_2O_3$) is interestingly low. As a result, most of the studied grains fall into the edenite category (Table 5).

**Table 5.** Composition of amphibole from inclusions in chromite grains of Almaz-Zhemchuzhina and Geofizicheskoe-VII deposits.

| № п/п | Almaz-Zhemchuzhina | | | | | | | Geofizicheskoe-VII | | | | | | | |
|---|---|---|---|---|---|---|---|---|---|---|---|---|---|---|---|
| wt.% | Chromio-Edenite | | | Chromio-Tremolite | | | Mhb | Chromio-Edenite | | | | | | | Prg |
| $SiO_2$ | 49.60 | 49.09 | 50.06 | 54.06 | 53.82 | 51.59 | 50.68 | 47.14 | 49.16 | 49.35 | 48.81 | 46.36 | 46.76 | 47.71 | 46.33 |
| $TiO_2$ | 0.41 | 0.22 | 0.33 | 0.37 | 0.24 | 0.56 | 0.47 | 0.30 | 0.38 | 0.25 | 0.37 | 0.32 | 0.43 | 0.32 | 0.48 |
| $Cr_2O_3$ | 2.92 | 2.63 | 2.72 | 1.77 | 1.56 | 2.60 | 2.62 | 3.17 | 2.76 | 2.73 | 2.76 | 3.16 | 3.08 | 2.83 | 1.89 |
| $Al_2O_3$ | 7.45 | 7.54 | 6.95 | 3.65 | 3.58 | 4.75 | 6.21 | 8.69 | 6.93 | 6.97 | 7.33 | 9.06 | 9.64 | 7.43 | 12.92 |
| FeO | 1.35 | 1.11 | 1.23 | 1.47 | 1.02 | 1.03 | 1.36 | 0.95 | 0.99 | 0.90 | 1.00 | 1.21 | 1.58 | 1.11 | 2.84 |
| MgO | 21.63 | 22.04 | 21.64 | 23.43 | 22.58 | 22.14 | 21.68 | 22.77 | 22.31 | 22.37 | 22.29 | 21.07 | 21.02 | 21.34 | 19.42 |
| NiO | bdl | bdl | bdl | bdl | 0.29 | 0.33 | 0.24 | bdl | bdl | bdl | bdl | 0.23 | bdl | 0.20 | bdl |
| CaO | 13.00 | 12.78 | 13.38 | 11.63 | 13.24 | 11.76 | 11.24 | 12.21 | 12.17 | 12.56 | 12.29 | 13.04 | 12.24 | 12.72 | 12.52 |
| $Na_2O$ | 2.35 | 2.72 | 2.10 | 1.05 | 0.77 | 1.63 | 1.63 | 3.25 | 2.57 | 2.67 | 2.66 | 3.14 | 3.13 | 2.43 | 2.10 |
| $K_2O$ | 0.00 | 0.10 | bdl | 0.11 | 0.25 | bdl | bdl | 0.17 | 0.19 | bdl | 0.19 | 0.14 | 0.12 | 0.09 | bdl |
| Total | 98.71 | 98.24 | 98.42 | 97.54 | 97.35 | 96.38 | 96.13 | 98.65 | 97.46 | 97.80 | 97.70 | 97.74 | 98.01 | 96.17 | 98.50 |
| apfu | | | | | | | | | | | | | | | |
| Si | 6.90 | 6.84 | 6.97 | 7.48 | 7.48 | 7.26 | 7.15 | 6.57 | 6.89 | 6.89 | 6.83 | 6.56 | 6.57 | 6.79 | 6.45 |
| Al(IV) | 1.06 | 1.12 | 0.99 | 0.48 | 0.47 | 0.70 | 0.82 | 1.40 | 1.07 | 1.08 | 1.13 | 1.41 | 1.40 | 1.19 | 1.54 |
| Al(VI) | 0.16 | 0.11 | 0.15 | 0.11 | 0.11 | 0.09 | 0.21 | 0.03 | 0.07 | 0.07 | 0.08 | 0.10 | 0.20 | 0.06 | 0.58 |
| Ti | 0.04 | 0.02 | 0.03 | 0.04 | 0.03 | 0.05 | 0.04 | 0.03 | 0.04 | 0.03 | 0.04 | 0.03 | 0.04 | 0.03 | 0.02 |
| Cr | 0.32 | 0.29 | 0.30 | 0.19 | 0.17 | 0.29 | 0.29 | 0.35 | 0.31 | 0.30 | 0.31 | 0.35 | 0.34 | 0.32 | 0.21 |
| Fe | 0.33 | 0.22 | 0.36 | 0.17 | 0.26 | 0.22 | 0.17 | 0.11 | 0.15 | 0.17 | 0.14 | 0.33 | 0.25 | 0.29 | 0.33 |
| Mg | 4.49 | 4.58 | 4.49 | 4.83 | 4.68 | 4.65 | 4.56 | 4.73 | 4.66 | 4.66 | 4.65 | 4.44 | 4.40 | 4.53 | 4.03 |
| Ni | | | | | 0.03 | 0.04 | 0.03 | | | | | 0.03 | | 0.02 | |
| Ca | 1.94 | 1.91 | 2.00 | 1.72 | 1.98 | 1.77 | 1.70 | 1.82 | 1.83 | 1.88 | 1.84 | 1.98 | 1.84 | 1.94 | 1.87 |
| Na | 0.63 | 0.73 | 0.57 | 0.28 | 0.21 | 0.45 | 0.45 | 0.88 | 0.70 | 0.72 | 0.72 | 0.86 | 0.85 | 0.67 | 0.57 |
| K | 0.00 | 0.02 | | 0.01 | 0.04 | | | 0.03 | 0.04 | | 0.04 | 0.03 | 0.02 | 0.02 | |

SEM EDS data. bdl—below detection limit, Mhb—magnesio-hornblende, Prg—pargasite.

Alloys, sulfides, sulfoarsenides, Ni, Fe, Cu, and Co arsenides are quite common in the studied ore samples (Table S1). They often occur in interstices of grains in association with secondary minerals (serpentine, chlorite), and less often with amphibole. In chromitites of the studied deposits, sulfides and arsenides of the Ni–Fe–Co–Cu system are represented by small clusters both inside chromite grains and in cracks filled with serpentine. The size of the inclusions varies from a few micrometers to 15–25 μm. The most numerous grains are represented by heazlewoodite (Figure 6a–d), with which tiny grains of nickeline (Figure 6a,b), native copper (Figure 6c) and PGMs (Figure 6d) can be associated. Pentlandite is much less common, and, in rare cases, cobalt-bearing pentlandite is noted (Figure 6e,f). In some grains of pentlandite, rather high contents of arsenic and the finest inclusions of PGMs were revealed. Zonal intergrowths are noted fairly often (Figure 6e); their periphery is composed of awaruite, and their central part is composed of cobalt-bearing pentlandite. Reverse relationships were noted less often (Figure 6f). In addition, chromitite samples from the Almaz-Zhemchuzhina deposit contain tiny grains of millerite and chalcocite (15–25 μm). Some grains of millerite are characterized by the presence of copper impurities (up to 5.45–8.6 wt.%), while all studied heazlewoodite and millerite grains are characterized by low iron contents (<1 wt.%). In some grains of chalcocite, the concentration of iron is

increased (up to 10 wt.%). Native minerals are represented by copper with a high nickel content (up to 21 wt.%) and almost pure nickel.

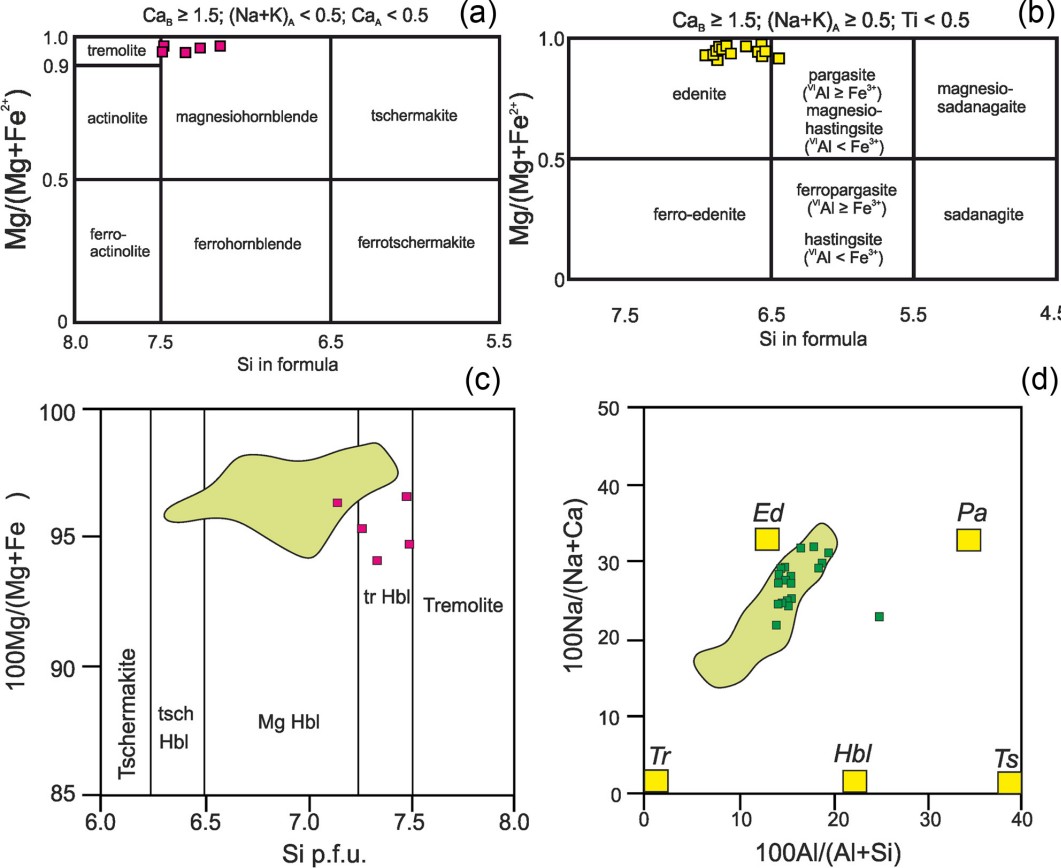

**Figure 5.** Composition of amphiboles from inclusions in Dzharlybutak chromite grains: (**a**,**b**) classification diagrams for amphiboles, following [42]; (**c**) 100 Mg/(Mg + Fe) vs. Si p.f.u. for amphiboles, following [32]; (**d**) 100 Na/(Na + Ca) vs. 100 Al/(Al + Si) for amphiboles, following [32]. Fields in (**c**,**d**) are compositions of amphibole grains from Kempirsai chromitite (Main Ore Field), following [32]; Ed—edenite, Hbl—hornblende; Pa—pargasite, Tr—tremolite, Ts and Tsch—Tschermakite.

PGMs in the chromitites from the South Kempirsai deposits were described in many earlier works [28,30,34,35]. In chromitites of the Dzharlybutak ore cluster, PGMs were found in all the studied samples. They were observed exclusively within chromite grains (Figure 7), and usually have very small sizes—from fractions of a micron to 3–5 μm, and rarely up to 10 μm. At the same time, one interesting feature of PGM inclusions in the chromite grains in all the studied samples is their close association with inclusions of hydroxyl-bearing silicate minerals, i.e., amphibole (Figure 7c–e), and rarely chlorite.

In the chromitite samples from the Geofizicheskoe-VII deposit, PGMs compositionally refer to disulfides of a laurite–erlichmanite series ($RuS_2$–$OsS_2$) and variable ratios between platinum group elements (Table 6). Iridium dominates in the composition of some grains, and their formula is close to $IrS_2$. Some inclusions are represented by PGE sulfoarsenides, which are compositionally close to irarsite. Solid solutions of PGEs with Ru–Os–Ir–Fe compositions were found in some chromite grains. Several PGM grains are associated with chlorite. Among other PGEs, laurite–erlichmanite sulfides contain a constant admixture of rhodium, in an amount of up to 3 wt.%; in iridium sulfoarsenides, its concentration increases to 5–6 wt.%. In single PGM grains, an admixture of platinum in the amount of 1–2 wt.% is noted, and one tiny grain of native copper containing 7 wt.% Pt was also revealed. Palladium was not found in the studied minerals.

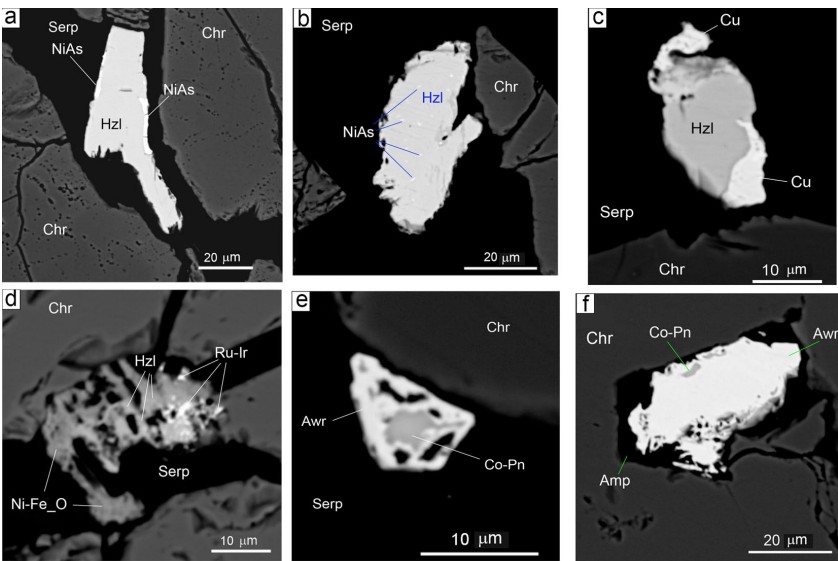

**Figure 6.** Minerals of base metals (Fe, Ni, Cu, Co) between and within the Cr-spinel grains from the Dzharlybutak group deposits. (**a**–**d**) Heazlewoodite grains in serpentine, filling cracks in a chromitite aggregate: (**a**) nickeline rim around heazlewoodite grains, (**b**) tiny inclusions of nickeline in heazlewoodite grains, (**c**) native copper in the periphery of the heazlewoodite grain, (**d**) partially oxidized heazlewoodite grain with tiny PGM inclusions in a chromite grain crack, (**e**) zoned intergrowth of Co-bearing pentlandite and awaruite in the serpentine matrix, (**f**) complex inclusion in the chromite grain that contains amphibole, Co-bearing pentlandite and awaruite. Amp—amphibole, Awr—awaruite, Co-Pn—Co-bearing pentlandite, Chr—chromite, Hzl—heazlewoodite, Serp—serpentine.

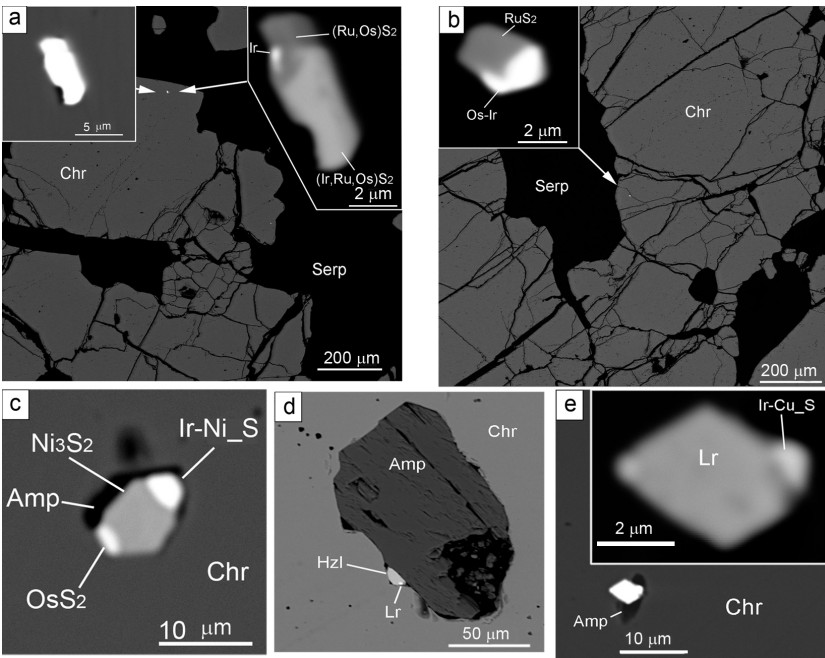

**Figure 7.** Localization of PGM in chromitites from the Dzharlybutak group deposits. (**a**) Intergrowth of PGE sulfides with a submicron inclusion of native iridium on the rim of a chromite grain, (**b**) intergrowth of laurite and Os-Ir alloy on the periphery of a chromite grain, (**c**–**e**) PGM inclusions in the central part of chromite grains: (**c**) intergrowth of heazlewoodite, erlichmanite, an Ir-Ni-S phase and amphibole, (**d**) intergrowth of heazlewoodite and laurite in association with amphibole inclusions in a chromite grain, (**e**) small laurite grain with a submicron Ir-Cu-S phase, associated with amphibole in a chromite grain. Amp—amphibole, Chr—chromite, Lr—laurite, Serp—serpentine.

**Table 6.** Composition of the PGE-bearing minerals from the Almaz-Zhemchuzhina and Geofizicheskoe-VII chromite ores, wt.% (SEM EDS data).

| Mineral | Dep. | S | Fe | Ni | Cu | As | Ru | Rh | Os | Ir | Total | Mineral Formula |
|---|---|---|---|---|---|---|---|---|---|---|---|---|
| Laurite | AZH | 31.80 | 0.85 | 0.46 | bdl | bdl | 27.78 | bdl | 23.97 | 9.62 | 94.48 | $(Ru_{0.56}Os_{0.25}Fe_{0.03}Ir_{0.10}Ni_{0.01})_{0.95}S_{2.02}$ |
| Laurite | AZH | 37.32 | 0.38 | bdl | bdl | 0.37 | 49.66 | bdl | 5.26 | 8.64 | 101.63 | $(Ru_{0.84}Ir_{0.07}Os_{0.04}Fe_{0.01})_{0.96}S_{2.00}$ |
| Laurite | AZH | 31.81 | 0.44 | bdl | bdl | bdl | 26.50 | bdl | 27.13 | 10.68 | 96.56 | $(Ru_{0.53}Os_{0.29}Ir_{0.11}Fe_{0.01})_{0.94}S_{2.03}$ |
| Laurite | AZH | 34.08 | 0.34 | bdl | bdl | bdl | 35.96 | bdl | 14.75 | 12.03 | 97.16 | $(Ru_{0.68}Os_{0.14}Ir_{0.11}Fe_{0.01})_{0.94}S_{2.03}$ |
| Laurite | AZH | 33.97 | 0.43 | bdl | bdl | bdl | 34.62 | bdl | 15.95 | 13.20 | 98.17 | $(Ru_{0.65}Os_{0.16}Ir_{0.13}Fe_{0.01})_{0.95}S_{2.03}$ |
| Laurite | AZH | 34.46 | 0.64 | bdl | bdl | bdl | 39.58 | bdl | 13.01 | 7.77 | 95.46 | $(Ru_{0.74}Os_{0.12}Ir_{0.07}Fe_{0.02})_{0.95}S_{2.03}$ |
| Laurite | GVII | 37.04 | 0.55 | bdl | bdl | bdl | 45.62 | bdl | 9.66 | 7.45 | 100.32 | $(Ru_{0.79}Os_{0.08}Ir_{0.06}Fe_{0.01})_{0.94}S_{2.03}$ |
| Laurite | GVII | 35.19 | 0.56 | bdl | bdl | 0.36 | 36.17 | bdl | 19.15 | 9.04 | 100.47 | $(Ru_{0.66}Os_{0.18}Ir_{0.08}Fe_{0.01})_{0.93}S_{2.03}$ |
| Laurite | GVII | 36.33 | bdl | bdl | bdl | 0.50 | 50.08 | 2.61 | 3.58 | 5.24 | 98.34 | $(Ru_{0.87}Ir_{0.04}Rh_{0.04}Os_{0.03})_{0.98}(S_{1.99}As_{0.01})_{2.00}$ |
| Laurite | GVII | 33.65 | 0.83 | bdl | bdl | 0.77 | 38.28 | bdl | 17.70 | 6.39 | 97.62 | $(Ru_{0.71}Os_{0.17}Ir_{0.06}Fe_{0.02})_{0.96}(S_{1.99}As_{0.01})_{2.00}$ |
| Laurite | GVII | 35.51 | 0.52 | bdl | bdl | bdl | 45.01 | bdl | 8.98 | 6.44 | 96.46 | $(Ru_{0.81}Os_{0.08}Ir_{0.06}Fe_{0.01})_{0.96}S_{2.02}$ |
| Laurite | GVII | 36.76 | bdl | bdl | bdl | bdl | 49.00 | bdl | 8.76 | 5.03 | 99.55 | $(Ru_{0.85}Os_{0.08}Ir_{0.04})_{0.97}S_{2.01}$ |
| Laurite | GVII | 36.48 | 0.55 | bdl | bdl | 0.54 | 50.90 | 2.17 | 3.72 | 4.53 | 98.89 | $(Ru_{0.87}Ir_{0.04}Rh_{0.03}Os_{0.03}Fe_{0.01})_{0.98}(S_{1.98}As_{0.01})_{1.99}$ |
| Laurite | GVII | 36.98 | bdl | bdl | bdl | bdl | 50.97 | 1.30 | 3.26 | 6.59 | 99.10 | $(Ru_{0.87}Ir_{0.05}Rh_{0.02}Os_{0.02})_{0.96}S_{2.00}$ |
| Laurite | GVII | 35.06 | 0.51 | bdl | bdl | bdl | 49.64 | bdl | 7.12 | 6.49 | 98.82 | $(Ru_{0.88}Os_{0.06}Ir_{0.06}Fe_{0.01})_{1.01}S_{1.97}$ |
| Laurite | GVII | 32.16 | bdl | bdl | bdl | 0.99 | 30.93 | 1.66 | 19.09 | 11.31 | 96.14 | $(Ru_{0.61}Ir_{0.11}Rh_{0.03}Os_{0.20})_{0.95}(S_{2.00}As_{0.02})_{2.02}$ |
| Erlichmanite | AZH | 29.30 | 1.00 | 0.88 | bdl | 1.06 | 12.62 | bdl | 38.36 | 13.38 | 96.60 | $(Os_{0.44}Ru_{0.27}Ir_{0.15}Fe_{0.03}Ni_{0.03})_{0.92}(S_{2.02}As_{0.03})_{2.05}$ |
| Erlichmanite | AZH | 27.52 | 0.84 | 0.47 | bdl | 0.63 | 6.98 | bdl | 45.79 | 12.87 | 95.10 | $(Os_{0.57}Ru_{0.16}Ir_{0.15}Fe_{0.03}Ni_{0.01})_{0.92}(S_{2.03}As_{0.01})_{2.04}$ |
| Erlichmanite | GVII | 29.35 | 0.83 | bdl | bdl | 1.02 | 13.30 | bdl | 39.92 | 11.47 | 95.89 | $(Os_{0.46}Ru_{0.29}Ir_{0.13}Fe_{0.03})_{0.91}(S_{2.04}As_{0.03})_{2.07}$ |
| Cuproiridsite | AZH | 28.57 | 1.34 | 1.57 | 9.34 | bdl | bdl | 6.42 | bdl | 52.78 | 100.00 | $(Cu_{0.72}Ni_{0.13}Fe_{0.11})_{0.96}(Ir_{1.34}Rh_{0.30})_{1.64}S_{4.37}$ |
| Cuproiridsite | GVII | 24.53 | 1.41 | 2.12 | 9.45 | bdl | bdl | 3.23 | bdl | 59.17 | 99.91 | $(Cu_{0.79}Ni_{0.19}Fe_{0.13})_{1.11}(Ir_{1.63}Rh_{0.16})_{1.79}S_{4.07}$ |
| Kashinite | AZH | 24.08 | 5.47 | bdl | 5.04 | bdl | bdl | 1.24 | bdl | 63.73 | 99.56 | $(Ir_{1.30}Fe_{0.38}Cu_{0.31}Rh_{0.04})_{2.03}S_{2.95}$ |
| Kashinite | GVII | 26.23 | 0.82 | 5.07 | 5.93 | bdl | 2.93 | 10.70 | bdl | 47.47 | 99.15 | $(Ir_{0.88}Rh_{0.37}Cu_{0.33}Ni_{0.31}Fe_{0.05}Ru_{0.01})_{1.95}S_{2.93}$ |
| Irarsite | AZH | 11.26 | 0.56 | 1.10 | bdl | 28.19 | bdl | bdl | bdl | 56.28 | 97.39 | $(Ir_{0.83}Ni_{0.05}Fe_{0.02})_{0.90}As_{1.07}S_{1.00}$ |
| Irarsite | AZH | 13.95 | 1.23 | 0.57 | bdl | 21.21 | 1.19 | 1.69 | bdl | 56.02 | 95.86 | $(Ir_{0.81}Fe_{0.06}Rh_{0.04}Ru_{0.03}Ni_{0.02})_{0.96}As_{0.79}S_{1.22}$ |
| Irarsite | AZH | 15.32 | 0.65 | 0.25 | bdl | 19.32 | 2.22 | 0.87 | 6.96 | 49.05 | 94.64 | $(Ir_{0.71}Os_{0.10}Ru_{0.06}Fe_{0.03}Rh_{0.02}Ni_{0.01})_{0.93}As_{0.72}S_{1.33}$ |
| Irarsite | GVII | 13.10 | 1.47 | bdl | 4.90 | 24.12 | 1.26 | 0.96 | 3.44 | 46.93 | 96.18 | $(Ir_{0.65}Cu_{0.20}Fe_{0.07}Os_{0.04}Ru_{0.03}Rh_{0.02})_{1.01}As_{0.86}S_{1.09}$ |
| Irarsite | GVII | 11.50 | 1.97 | bdl | 3.35 | 21.97 | 1.12 | 0.67 | 3.36 | 49.38 | 93.32 | $(Ir_{0.74}Cu_{0.15}Os_{0.05}Ru_{0.03}Fe_{0.10}Rh_{0.01})_{1.08}As_{0.85}S_{1.04}$ |
| Osarsite | AZH | 11.13 | 0.71 | bdl | bdl | 30.39 | 6.93 | bdl | 47.63 | 3.19 | 99.98 | $(Os_{0.68}Ru_{0.18}Ir_{0.04}Fe_{0.03})_{0.93}As_{1.10}S_{0.94}$ |
| Ruarsite | GVII | 12.85 | 0.40 | bdl | bdl | 33.25 | 35.65 | bdl | 6.53 | 4.59 | 93.27 | $(Ru_{0.83}Os_{0.08}Ir_{0.05}Fe_{0.01})_{0.97}As_{1.05}S_{0.95}$ |
| Ruarsite | GVII | 12.82 | 0.36 | bdl | bdl | 33.17 | 34.36 | bdl | 8.00 | 4.91 | 93.62 | $(Ru_{0.81}Ir_{0.06}Os_{0.10}Fe_{0.01})_{0.98}As_{1.05}S_{0.95}$ |
| Iridium | AZH | bdl | 0.33 | bdl | bdl | bdl | 10.70 | bdl | 35.59 | 45.86 | 92.48 | $Ir_{0.44}Os_{0.34}Ru_{0.19}Fe_{0.01}$ |
| Iridium | AZH | bdl | 0.37 | bdl | bdl | bdl | 10.61 | bdl | 36.16 | 45.44 | 92.58 | $Ir_{0.43}Os_{0.35}Ru_{0.19}Fe_{0.01}$ |
| Iridium | AZH | bdl | 0.60 | bdl | bdl | bdl | 3.19 | bdl | 41.74 | 47.27 | 92.80 | $Ir_{0.48}Os_{0.43}Ru_{0.06}Fe_{0.02}$ |
| Iridium | GVII | bdl | 0.67 | bdl | bdl | bdl | 3.12 | bdl | 41.90 | 47.59 | 93.28 | $Ir_{0.48}Os_{0.43}Ru_{0.06}Fe_{0.02}$ |
| Osmium | GVII | bdl | 0.85 | bdl | bdl | bdl | 2.87 | bdl | 46.58 | 42.68 | 92.98 | $Os_{0.47}Ir_{0.43}Ru_{0.05}Fe_{0.02}$ |
| Osmium | GVII | bdl | 0.97 | bdl | bdl | bdl | 3.50 | bdl | 47.26 | 42.60 | 94.33 | $Os_{0.47}Ir_{0.42}Ru_{0.06}Fe_{0.03}$ |

AZH—Almaz-Zhemchuzhina deposit, bdl—below detection limit, GVII—Geofizicheskoe-VII deposit.

In the chromitite samples from the Almaz-Zhemchuzhina deposit, all PGM inclusions can be divided into homogeneous (Figure 8) and polymineral (Figure 9). Homogeneous grains predominate (60%). Compositionally, they refer to disulfides of the laurite–erlichmanite series, with variable ratios between PGE. Iridium plays the leading role in the development of the polymineral intergrowths of PGM. Iridium occurs as the following phases: native, osmium iridium, and an essentially iridium sulfide variety, which has a formula close to kashinite and iridisite (15%). Sulfoarsenides are represented by, essentially, both iridium-irarsite (about 10%) and osmium-osarsite (about 10%) varieties (Table 6). Sulfides of complex Ni-Cu-PGE composition including cuproiridsite (less than 10%) with variable ratios of metals are observed in approximately comparable amounts.

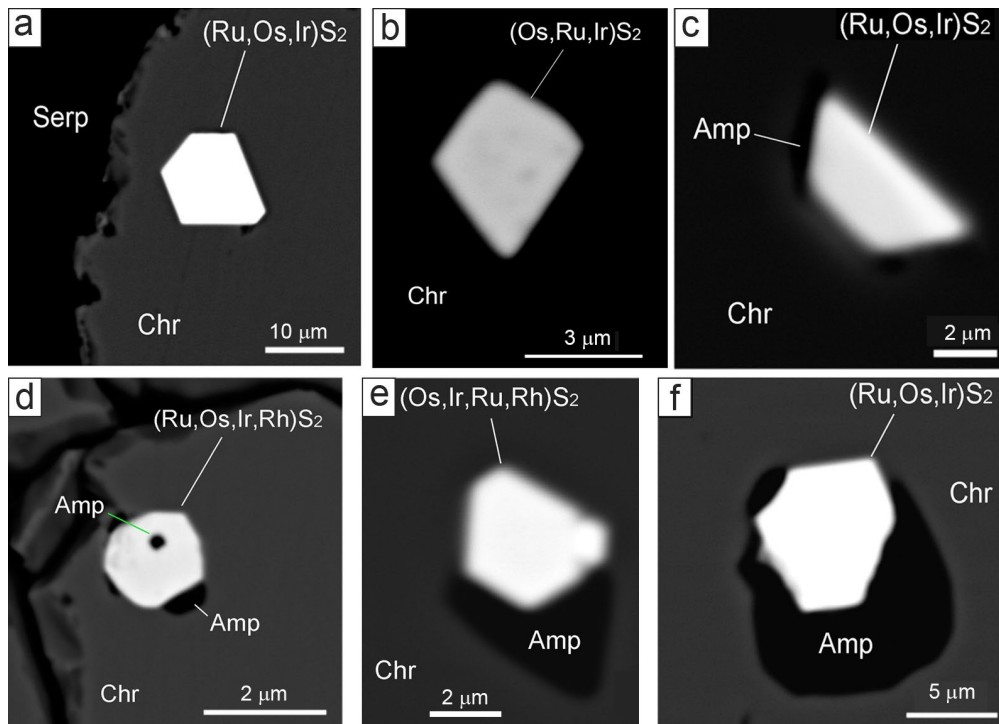

**Figure 8.** Euhedral grains of laurite–erlichmanite disulfides in chromitites from the Dzharlybutak group deposits. (**a**,**b**) PGE sulfide inclusions in chromite grains, (**c**) laurite inclusion with a fine rim of amphibole in a chromite grain, (**d**) complex inclusion of laurite and amphibole on the periphery of a chromite grain, (**e**,**f**) laurite–erlichmanite sulfides tightly associated with amphibole grains in a chromite grain. Amp—amphibole, Chr—chromite, Serp—serpentine.

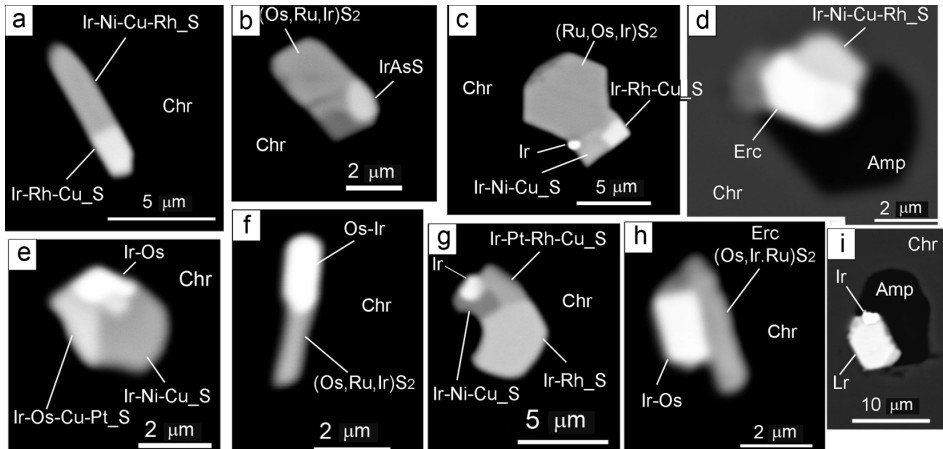

**Figure 9.** Morphology and internal structure of PGM intergrowths in chromitites from the Dzharlybutak group deposits. (**a**) intergrowth of Ir-Ni-Cu-Rh-S and Ir-Rh-Cu-S phases of rod-like morphology in the chromite, (**b**) intergrowth of erlichmanite and irarsite in a chromite grain, (**c**) intergrowth of complex composition (laurite, native Ir, Ir-Ni-Cu-S and Ir-Rh-Cu-S phases) in a chromite grain, (**d**) intergrowth of erlichmanite and Ir-Ni-Cu-Rh sulfide with amphibole in chromite, (**e**) intergrowth of Ir-Os alloy with Ir-Os-Cu-Pt and Ir-Ni-Cu sulfides in a chromite grain, (**f**) rod-like intergrowth of Ir-Os alloy and erlichmanite in chromite, (**g**) four-phase intergrowth of Ir-Ni-Cu, Ir-Rh and Ir-Pt-Rh-Cu sulfides with native Ir in a chromite grain, (**h**) intergrowth of erlichmanite and Ir-Os alloy in chromite, (**i**) intergrowth of laurite and native Ir with amphibole in a chromite grain. Chr—chromite, Erc—erlichmanite, Lr—laurite.

Mineral inclusions of the laurite–erlichmanite series typically show a high degree of idiomorphism (Figure 8). PGM grains are often captured in "amphibole traps" (Figure 8c–e), with different areal ratios. Sometimes they contain submicron inclusions of this mineral (Figure 8d). In the structure of PGM polymineral clusters, minerals of the laurite–erlichmanite series are also common and have the greatest influence (Figure 9b–d,f,h,i), while inclusions of sulfides, sulfoarsenides and native iridium (Os-Ir or Ir-Os phases) are subordinate (Table 6). In some cases, intergrowths are only represented by iridium minerals of varied composition (Figure 9a,e,g). Polymineral intergrowths of PGM can be tightly associated with amphibole as well (Figure 9d,i).

In addition to typical minerals of ultramafic rocks and chromitites, in the studied polished sections, we found single grains of minerals considered "exotic" for these associations, i.e., titanite, zircon, monazite, apatite, and a Ca-Ti-O phase (presumably, kassite). In order to exclude "artifacts", in Figure 10 we present pairs of images in different modes—secondary electrons (SE) and back-scattered electrons (BSE). Single grains of zircon, monazite, and barite were found in chromitite samples from the Almaz-Zhemchuzhina deposit in interstices with serpentine (Figure 10a–f). Phlogopite rarely occurs as primary inclusions in chromite grains; it is also more often clustered in interstices, in association with the secondary minerals (Figure 10g–i).

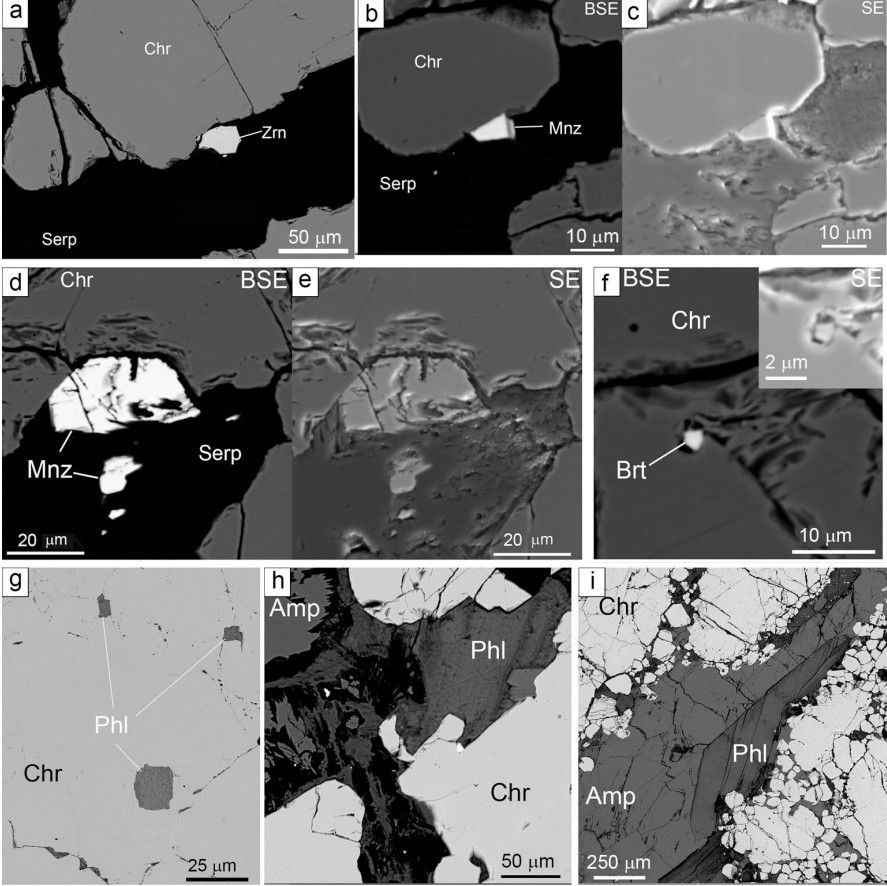

**Figure 10.** Exotic accessory minerals in chromitites from the Dzharlybutak group deposits. (**a**) Small zircon grain in a crack filled by serpentine at the boundary with a chromite grain, (**b**–**e**) euhedral monazite grains in a crack filled by serpentine at the boundary with a chromite grain, (**f**) barite inclusion in a chromite grain, (**g**) phlogopite inclusions in a chromite grain, (**h,i**) phlogopite and amphibole filling cracks in chromitite. Amp—amphibole, Brt—barite, Chr—chromite, Mnz—monazite, Phl—phlogopite, Serp—serpentine, Zrn—zircon.

Apatite was found both as a primary inclusion, together with heazlewoodite in chromite, and in association with amphibole, in interstices between ore grains. Titanite was mainly observed in interstices between chromite grains or near cracks, in association with veined amphibole and chlorite. The Ca–Ti–O mineral we found in the silicate inclusions in chromite (associated with serpentine and chlorite) (Figure 11) is not perovskite, since analyses show low amounts, most likely indicates a high hydroxyl content. The estimation demonstrates its compositional proximity to kassite or cafetite. Kassite was previously described in the chromitites of the Saranovsky deposit [43].

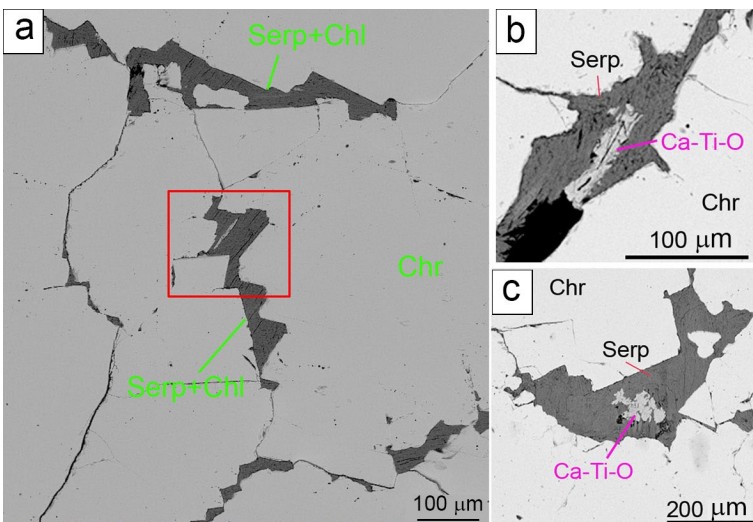

**Figure 11.** Ca-Ti-O composition in chromitites from the Almaz-Zhemchuzhina deposit: (**a**) general view, (**b**,**c**) detailed images of the Ca-Ti-O mineral phase. Chl—chlorite, Chr—chromite, Serp—serpentine.

## 5. Discussion

The set of minerals in the interstices of ores and inclusions inside chromite grains of the Dzharlybutak ore cluster deposits is generally similar to that of previously studied deposits of the ophiolite type. At the same time, the conducted studies revealed specific compositional features of PGMs, olivine with abnormally high nickel concentrations, as well as the release of rare minerals in ultramafic rocks (zircon, monazite, etc.) in situ, and not only in concentrates.

Minerals such as olivine, serpentine, orthopyroxene, and chlorite are typical of primary or altered ultramafic rocks and their presence is easily interpreted. In particular, the presence of rounded olivine grains in Cr-spinels can be explained by their capture during chromite crystallization, either from melts percolating through restite [9] or during syndeformational growth [26].

The studied chromitites invariably have been observed to be associated with exclusively magnesian olivine and high-Cr chromite, as can be seen in the OSMA diagram (Figure 12a). In order to determine the formation settings of the primary mineral associations of chromitites, we calculated the closure temperatures of exchange reactions in olivine–chromium spinel pairs using various versions of geothermometers [44–46], as well as oxygen fugacity for the same pairs of minerals, according to the oxybarometer results from [44].

Previously, numerous estimates for the olivine–Cr-spinel pair from samples of the Kempirsai massif were carried out in several works [47–50]. It was found that in the chromitites of the mantle section, the temperatures of mineral equilibrium range from 1200 to 600 °C, and the oxygen fugacity ranges from 1.7 to + 2.73 ΔFMQ. The data obtained in this study showed that the formation of chromitite bodies occurred at subsolidus temperatures of ultramafic rocks (700–850 °C) and an oxygen fugacity of −1.04 to +2.8 ΔFMQ (Figure 12b),

which is comparable with previous estimates for both Kempirsai chromitites [47–49] and peridotites from the more northern regions of the Southern Urals [11].

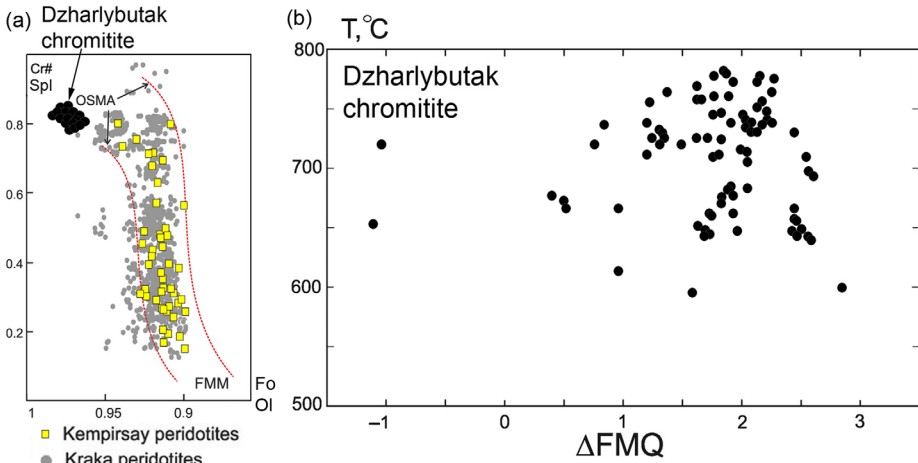

**Figure 12.** (**a**) OSMA diagram for coexisting olivine and Cr-spinel grains; (**b**) $\Delta\log(fO_2)$ vs. T diagram for olivine inclusions in chromite grains. Here, OSMA is "olivine-spinel mantle association" and FMM is "fertile mantle material", which corresponds to Ol-Spl compositions in restite; Ol-Spl compositions are found in in lherzolite-type massifs of the South Urals and peridotite from the Almaz-Zhemchuzhina deposit area, as reported in [27,51].

These data, together with other features of the composition of olivine and chromite, allow researchers to interpret them in different ways. In particular, in works of Chaschukhin I.S., the idea of the formation of chromite mineralization as a result of the removal of chromium by reduced fluids from the deep levels of the lherzolite section is promoted [48,49], and in the works of other authors, the primary magmatic genesis of the olivine–Cr-spinel association is suggested [50]. However, we believe that the anomalously high content of nickel and #Mg in olivine indicates an extremely depleted composition, which tests the idea of a restite origin of ultramafic rocks and chromitites. The data obtained also do not contradict the formation of chromitites in the upper mantle of a fore-arc basin, as suggested in [32,52].

Numerous inclusions of amphibole and phlogopite in chromite are commonly interpreted as a result of either the fluid–metasomatic genesis of chromitites [53], or the reaction of restite with percolating melts and fluids [19,20], known as "mantle metasomatism" [35]. The main argument here is the very presence of hydroxyl-bearing phases inside unaltered chromite grains. However, in our previous works, we used numerous examples to show the co-existence of the finest inclusions of Cr-spinel and amphibole inside unaltered olivine and orthopyroxene crystals from lherzolites [54]. This may testify to the solid-phase nature of chromite crystallization, with inclusions generated by multiple recrystallizations inside the plastically deformed mantle material.

The PGM mineralization in the studied chromitites is characterized by the prevalence of refractory platinoids (ruthenium, iridium and osmium), a subordinate role of rhodium and platinum, and the complete absence of palladium. This also complies with formerly obtained data on some deposits in the south-eastern part of the Kempirsai massif [30,32]. Noteworthy, we found no solid solutions with the Rh–Zn–Pt composition, which was reported in [36] for the section of the Almaz-Zhemchuzhina deposit. This is more likely due to the zonal distribution of PGMs of varied compositions at the deposit, and the absence of the above-mentioned phase in the interval of 750–1100 m, which we studied.

The vast majority of the studied PGM grains in chromites is represented by sulfides, while alloys comprise a minor proportion (Figure 13). In contrast to the previously studied chromitites from more northern regions of the Southern Urals (Kraka, Nurali), where laurite predominates [55], the composition of sulfides varies significantly and shows an almost

continuous series between essentially ruthenium (laurite) and osmium (erlichmanite) varieties. Interestingly, the role of iridium in the composition of sulfides is rather important. Among native PGM, iridium and an Os–Ir alloy prevail, while proportions of metals are approximately equal, and ruthenium is extremely rare. This fact also distinguishes the studied deposits from those of the Kraka massif.

In order to explain the genesis of the PGM inclusions in chromitites of podiform deposits, several different mechanisms have been proposed in previous works, depending on the location, morphology, and composition of the inclusions. One of the most comprehensive reviews on this issue [56] provides a classification of PGM, according to which all the minerals that we studied fall into type I (enriched with IPGE), and occur in the inner parts of chromite grains, with no visible connection to cracks (Subtype 1). The main hypotheses to explain the genesis of these inclusions are usually proposed as follows: (1) the incorporation of refractory platinoids into chromite at high mantle temperatures and their separation into their own phases upon cooling [57–59]; (2) simultaneous crystallization with chromite from melts or during the melt + peridotite reaction [60,61]; (3) crystallization, resulting from fluids and/or melts percolating through ultramafic rocks [35,36], including "supercritical fluids" [30].

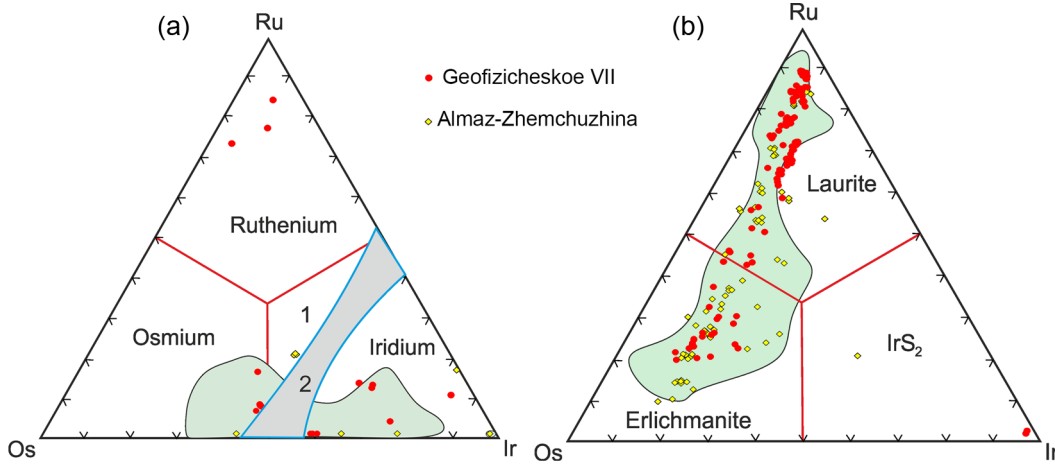

**Figure 13.** Compositions of the PGM alloys and sulfides in chromitites from the Almaz-Zhemchuzhina and Geofizicheskoe-VII deposits (**a**) Os-Ru-Ir alloys (at.%); (**b**) Os-Ru-Ir sulfides (at.%); (**a**) fields: 1—rutheniridosmine, 2—immiscibility area, following [62]; green fields indicate the composition of PGM from Kempirsai chromitite, following [32].

We must admit that the issue of the PGM inclusions genesis in chromite of podiform deposits makes constructing a simple model quite a challenge. At the same time, the grains we studied show a number of similar features that suggest the leading role of subsolidus processes in their genesis. First, chromitites always display elevated concentrations of platinoids that keep growing from disseminated ores to massive ones. This fact is not consistent with "magmatic" models that release chromite and PGM from basaltoid melts [56]. On the contrary, it supports the first hypothesis, regarding the "preexisting elevated concentrations" of PGM in mantle chromite [59]. Second, PGM inclusions are predominantly enclosed within chromite grains, which contradicts the "fluid" genesis of mineralization. Third, objections to the solid-phase genesis due to the association of PGMs with hydroxyl-bearing phases disappear once we assume that these associations are caused by the process of "impurity segregation" during the syntectonic recrystallization of chromite. All the "exotic" mineral inclusions in the studied chromitite samples (zircon, barite, monazite, kassite) are clustered in serpentine, and less often in a chlorite filling of the interstitial space, which most likely indicates their low-temperature genesis, simultaneously, with the indicated secondary minerals.

It should be noted that PGM and BMS are important indicators of sulfur fugacity in the magma process and chromitite recrystallization. The absence of a monosulfide solid solution in chromitites and, in general, the very rare occurrence of BMS, indicates very low values of $f$(S), and, consequently, the impossibility of the partitioning of metals into the sulfide melt. This leads to the frequent spread and formation of alloys. Minerals enriched in elements with low-melting points (such as, Bi, Sb, Te), which have a large distribution coefficient to the sulfide liquid, are practically not found in chromitites. In the samples we have studied, PGE disulfides of the laurite–erlichmanite series usually coexist with low-ruthenium Ir–Os alloys. This means that during the formation of these intergrowths, the sulfur fugacity was low, since the incorporation of osmium into laurite indicates an increase in sulfur fugacity during crystallization [63]. Such features have been noted both for the Ural deposits (including the Kempirsai Main Ore Field) and for deposits around the world [32,59,64].

## 6. Conclusions

The study of chromitites from the Dzharlybutak ore cluster deposits determined the following accessory minerals in inclusions from chromite grains, in addition to the interstitial secondary minerals (serpentine, chlorite): (1) nominally anhydrous silicates (olivine, orthopyroxene, clinopyroxene), (2) hydrous silicates (amphibole, phlogopite), (3) base metal sulfides, (4) PGM. The interstices contain sulfides, arsenides, native minerals, such as Fe, Cu, Ni, Co, as well as such "exotic" minerals rarely seen in ultramafic rocks and chromitite, including apatite, monazite, zircon, titanite, barite, and kassite.

Chromitites and their host dunites occur as restite, which is subject to transformation and rheomorphic differentiation within the mantle diapir. An intense depletion is evidenced from the magnesian (Fo$_{95-98}$) and anomalously nickel-rich (up to 1.8 wt.% NiO) composition of olivine inclusions. Based on the composition of coexisting olivine and chromite, the temperature conditions (700–850 °C) and oxygen fugacity (−1.04 to +2.8 ΔFMQ) at the closing of this exchange reaction were estimated. They most likely indicate the completion of high-temperature processes in the upper mantle settings of the fore-arc basin, which is consistent with the findings of previous researchers [32].

We connected the formation of inclusions of hydroxyl-bearing minerals (amphibole, phlogopite) in the cores of chromite grains with their capture at the early stages of the formation of Cr-spinel, caused by the decomposition of deformed pyroxenes in lherzolites and harzburgites [65]. The subsequent formation of chromitites followed the mechanism of rheomorphic differentiation within a localized zone of plastic flow (dunite). As a result, the inclusions were preserved inside a rigid chromite container.

The PGM inclusions in the inner parts of chromite grains were most likely formed under subsolidus conditions by IPGE being incorporated in the Cr-spinel lattice at high mantle temperatures, and then being segregated near defects and/or inclusions as a result of cooling, plastic deformation, and recrystallization of the host mineral. Interstitial sulfides, sulfoarsenide alloys of base metals and some "exotic" minerals (i.e., monazite, apatite, barite) were produced by the hydrothermal alteration of primary ultramafic rocks and chromitite.

**Supplementary Materials:** The following supporting information can be downloaded at: https://www.mdpi.com/article/10.3390/min13020263/s1, Table S1: Composition of PGM and base metal sulfides in chromitites of Dzharlybutak ore group.

**Author Contributions:** Conceptualization, D.E.S.; investigation, field work, sample preparation, D.E.S., D.K.M. and R.A.G.; writing—original draft preparation, D.E.S. and R.A.G.; writing—review and editing, A.V.V. and D.E.S.; validation—A.V.V.; data curation –A.V.V. and D.E.S.; visualization, D.E.S. and R.A.G.; project administration, D.E.S.; funding acquisition, D.E.S., A.V.V. and R.A.G. All authors have read and agreed to the published version of the manuscript.

**Funding:** This study was supported by the Russian Science Foundation grant no. 22-17-00019 (https://rscf.ru/project/22-17-00019/), accessed on 11 May 2022.

**Data Availability Statement:** The data presented in this study are available on request from the corresponding author.

**Acknowledgments:** We are very grateful to our colleagues from the "ERG Exploration" Ichshenko A.M., Khamzin A.B., Makatov Dastan K., Ulukpanov K.T., Alimov S. for their very sincere helps during the field trips to the Main Ore field deposits. Also, we are grateful to T.A. Miroshnichenko for the translation of the manuscript from Russian to English.

**Conflicts of Interest:** The authors declare no conflict of interest.

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
