# Peer review of "Accessory Minerals in the Chromitite Ores of Dzharlybutak Ore Group of Kempirsai Massif (Southern Urals, Kazakhstan): Clues for Ore Genesis"

_minerals, doi:10.3390/min13020263_

Round 1
Reviewer 1 Report
The article is interesting and original, it is devoted to the mineralogy of chromitites of two Kimpersai deposits - a giant chromite knot: one large and one small. The brevity of the substantiation of the genetic concept and the lack of explanation of the differences between two nearby deposits are confusing. There are a number of other comments.
In the introduction, with a general description of Kimpersai, there is not a single reference
L. 50 - inclusions in chromitite chrome spinels also contain more exotic minerals
L. 59 - what is the uniqueness of the deposit? In size or composition?
L. 71 - how, when shooting with an electron microscope, not qualitative, but quantitative analyzes were obtained?
Fig. 1 - there is no reference in the text, there is no inset of the geographical location of the array on the map of at least Kazakhstan, very small letter designations on the map, decoding of color designations should not be in the caption to the figure, as they do in Russia, but next to the diagram - next to the rectangles instead of numbers. Different symbols on the diagram and on the sidebar are also not very successful.
L. 90-95 - the relationship of ore fields, zones, clusters is not clear
Table 1 - 3 mineral - a mistake in the name? 4 mineral - can you clarify which amphibole? And the text does not further explain which minerals calcium amphiboles from inclusions belong to.
L. 140 - it is said about the limit of detection, but what it is unclear
Tables of mineral compositions take up too much space, it is better to give either average compositions with deviations or minimum and maximum values, and translate the tables themselves into Supplementary Materials in a smaller font
L. 412-413 - it is not clear why the presence of platinum minerals in the form of inclusions in chrome spinel contradicts their origin in connection with fluids, especially since amphibole is usually present in these inclusions. But what if the origin of chromitites is associated with fluid processes?
Bibliography: publications 4, 11, 25, 27 in Russian, but this is not indicated. We need to check - 11 could have been published in Geology of Ore Deposits
Author Response
"Please see the attachment."

Reviewer 2 Report
This paper reports a mineralogical investigation of podiform chromitites collected in the Kempirsai Ophiolite complex located in the Urals. The Kempirsai chromitites have been intensively studied in the past and the results reported in this paper do not represent a novelty. Furthermore several relevant references on the genesis of podiform chromitites and the accessory minerals are missing. The mineralogical data are poorly presented. In particular, no compositional diagrams are shown for chromite, olivine, amphibole and pyroxenes as well as the graphic showing the variation of the oxygen fugacity as a function of temperature. Tables are not readable. The chapter of methodology needs to report more precise information about the electron microprobe conditions, i.e. WDS or EDS, standards used, etc. etc..
What does it mean PGM segregation, a term very often used along the text?
Line 37 what is chloritolites?
Lines 45-53. This part of the paper must be totally rewritten citing the appropriate bibliography and do not mix Alaskan-type chromitites with the podiform one.
Line 49 PGM or PMGs? Please explain the meaning of the achronym.
Line 50 …..there are quite many “exotic” inclusions, i.e. amphiboles, phlogopite,…. Actually amphibole and phlogopite are found very often enclosed in chromite in podiform chromitites and cannot be considered exotic. Please cite the references.
Chapter 3. Results. This chapter reports geological information mixed with mineralogical results.
Line 264 ….laurite-erlichmanite series with the general formula (Ru,Os,Ir)S2….. This formula is those of laurite.
Table 8. Please explain why the totals are very low. What does mean formula coefficients?
In the chapter discussion the authors must include more specific citations.
The chapter conclusion is not clear.
For all these reasons I do not recommend the acceptance of the manuscript in this present form.
Author Response
"Please see the attachment."

Reviewer 3 Report
Please look at the attached annotated pdf here to help improving the revised version of the manuscript you will submit.
Title of the paper needs some little modifications (Line 2-3),
Re-edit some parts of the manuscript, e.g. the abstract, keywords and introduction.
When you use the term “chromitites”, it is preferable to replace it with “chromitite ore”. Also, your ore is a podiform type (Line 35 and elsewhere) so you can also use it as a genetic term all over the text.
Try to separate your aims of study (objectives) from methods (Line 63-85).
Line 75: What about detection limits and precision of the SEM as a semi-quantitative tool?. In such a case, electron microprobe analyses (EMPA) are much more recommended.
Lines 76 and 78, as well as elsewhere: Use structural formulae instead of formulas. You mix Latin with English plurals here.
Caption of Fig. 1 (Line 113): The ultramafics are derived by default. Please replace this by "ophiolitic serpentinites". Lines 114-115: Delete predominantly in the three ultramafic lithologies.
Lines 120-121: Are you sure that this magnesite is supergene, i.e. after emplacement and being exposed?. Do you have any reference to support this, so if any please provide one or two. Please do not discard the formation of magnesite in the oceanic environment.
In Table 1, the chemical formula needs to be subscripted in some minerals. Please check carefully.
In Table 1 too, the mineral No. 28 or the unnamed Ni-Cu-Ir-S phase: Remove the 3 dots. I know you mean the possibility that other elements would be present but it is preferable not to predict. Mineral #18 is miss-spelled as chalcocite. It should be chalcocite, which has its name from its copper content.
Line 136: I recommend to replace chromian spinel here and in all parts of the text as Cr-spinel.
Line 154-155: You need to distinguish between Cr-spinel and chromite, either chemically or mineralogically.
Table 2 (Line 175), and others: All tables of the chemical composition show very bad editing. Please pay attention for the font, width of columns and digits especially the zeros. Also, oxides of elements must be subscripted.
Line 196: Amphibole inclusions occur in two modes not “positions”. You mention here about textural status.
Lines 205-206: In terms of structural mineralogy, what do you mean be permanent Na and Cr?.
Lines 224-225: You move from a descriptive term here to a genetic one. Why do you consider your millerite and chalcocite as "precipitate"?.
Fig. 238: For this figure and others, where are the abbreviation of mineral names following what are shown in every microphotograph of the ore minerals and gangues as well.
In the captions of Figs 6 and 7 (Lines 256 and259), use PGE-bearing sulphides and sulfarsenides.
Lines 271 and 299: While you are presenting details of the mineral chemistry, try to avoid some genetic terms such as precipitation and segregation.
Line 316: Sphene is an old name. Please use titanite instead.
In the discussion section (Line 401), caption of figure 9 showing possible soild solutions should be modified.
Lines 417-419, discussing the confinement of the exotic mineral inclusions to serpentine and not to chlorite, please keep in mind that chlorite also does as a retrograde phase.
Your reference list should be revised carefully. Please follow the style of journal as indicated in the instructions for authors.

Round 2
Reviewer 1 Report
The authors generally took into account the comments and suggestions. This does not fully apply to tables and the role of fluids in the formation of both inclusions in chrome spinels and chromitites in general. But this is the author's position, it is acceptable.
L. 63 - incorrectly formatted link
Table 1 - 4 mineral - amphibole, but what kind of mineral is not clear. This is explained in the text. Can you make a link?
L. 469 - the parenthesis is missing
The bibliography lists several articles from the journal Zapiski RMO (35? 36). Many articles from this journal are reproduced in English in the journal Geology of Ore Deposits - must be checked
Author Response
Response to Reviewer 1 Comments
We are very grateful to the Reviewer for a careful reading of our manuscript and useful comments. All the mistakes you mentioned have been corrected.
Point 1 L. 63 - incorrectly formatted link
Response 1: It is corrected
Point 2 Table 1 - 4 mineral - amphibole, but what kind of mineral is not clear. This is explained in the text. Can you make a link?
Response 2. We believe that the listing in Table 1 of the detailed amphiboles names nomenclature will lead to an unnecessary complication that does not carry much genetic information. In the same way, we do not specify the chlorites and apatites. This is justified by the fact that even the tremolite that we analyzed contains 1-1.5 wt. % Na2O and 4-6 wt. % Al2O3, which are close the field of edenite, a typical Na-Ca amphibole. The same compositions are close to analyzes that are formally assigned to magnesiohornblendite and pargasite. Thus, all studied amphiboles can be considered of Ca-Na specific.
Point 3: L. 469 - the parenthesis is missing
Response 3: There is no mistake here, this is how the fourth point is indicated.
Point 4: The bibliography lists several articles from the journal Zapiski RMO (35-36). Many articles from this journal are reproduced in English in the journal Geology of Ore Deposits - must be checked
Response 4: On this point, the following should be said. Issues of Geology of Ore Deposits in English have been published since 2006. The article (Yurichev et al., 2019) is not available on the journal website.

Reviewer 2 Report
Dear Editors, I have read the revised version of the manuscript. The Authors have done a lot of work to improve their manuscript. This is greatly appreciated and now I recommend the publication after major revision. I still see the following main problems, i.e. the novelty, the handle of the mineralogical data and the lack of the appropriate bibliography of previous work done on the same subject. Few specific comments are listed in the following.
1) Although I am not too familiar with the Kempirsai massif, I know that it hosts the biggest podiform chromite deposits in the world. I suggest to the Authors to specify if the chromitite studied in this work were investigated before by other Authors or if these data refer to mineralization never investigated before.
2) The mineralogical diagrams are still of poor quality. I recommend to use more useful diagrams (a lot of examples are available in the literature, including several from Kempirsai), and to add new diagrams to discriminate also pyroxenes and amphibole. Maybe a comparison with literature data of Kempirsai will be also useful.
3) Finally the sentence ……The closure of exchange reactions between olivine 19 С and in the oxygen fugacity range of 20°and chromite occurred in the temperature range of 700–850 FMQ, which most likely corresponds to the upper mantle settings of the fore-arc basin 21D-1.04 … +2.8. The Authors should discuss better this point. Detailed studies of chromite-olivine thermometry and oxygen barometry in chromitites of the Urals, including Kempirsai, have been published by several Authors, such as Chashchukhin et al. (1996, 2002, 2007), Pushkarev (2000), Pushkarev and Anikina (2002), Pushkarev et al. (2007), Garuti et al. (2013). In particular the Authors should explain why these values are attributed to a specific geological setting such as a fore-arc basin and not with the strong metasomatic event that have affected the Kempirsai massif reported by several Authors (see for example Melcher et al. 1994, 1997).
Author Response
Response to Reviewer 2 Comments
We are very grateful to the Reviewer for a careful reading of our manuscript and useful comments. All the mistakes you mentioned have been corrected.
Point 1: I still see the following main problems, i.e. the novelty,
Although I am not too familiar with the Kempirsai massif, I know that it hosts the biggest podiform chromite deposits in the world. I suggest to the Authors to specify if the chromitite studied in this work were investigated before by other Authors or if these data refer to mineralization never investigated before.
Response 1: We supplemented the text by inserting several sentences into the introduction explaining the novelty and goal of our research. Unfortunately, we cannot specify the exact intervals studied, as this information is a trade secret.
Point 2: … the handle of the mineralogical data and
The mineralogical diagrams are still of poor quality. I recommend to use more useful diagrams (a lot of examples are available in the literature, including several from Kempirsai), and to add new diagrams to discriminate also pyroxenes and amphibole. Maybe a comparison with literature data of Kempirsai will be also useful.
Response 2: We supplemented the manuscript with new diagrams. Where necessary, we have added the composition fields of the relevant minerals from previous work.
Point 3: … the lack of the appropriate bibliography of previous work done on the same subject.
Finally the sentence ……The closure of exchange reactions between olivine 19 С and in the oxygen fugacity range of 20°and chromite occurred in the temperature range of 700–850 FMQ, which most likely corresponds to the upper mantle settings of the fore-arc basin -1.04 … +2.8. The Authors should discuss better this point. Detailed studies of chromite-olivine thermometry and oxygen barometry in chromitites of the Urals, including Kempirsai, have been published by several Authors, such as Chashchukhin et al. (1996, 2002, 2007), Pushkarev (2000), Pushkarev and Anikina (2002), Pushkarev et al. (2007), Garuti et al. (2013).
Response 3: We thank the Reviewer for this remark. Unfortunately, the review does not contain exact references to the cited works, and we believe that in the new version of the manuscript we were able to reflect the contribution of the mentioned authors to the study of the chromite-olivine association of the Kempirsai massif. We did not include in the Discussion the papers that are devoted to the Ural-Alaskian type massifs (Pushkarev, 2000; Pushkarev, Anikina, 2002; Pushkarev et al., 2007; Chashchukhin et al., 2002), because in the previous review we were remarked precisely on this about.
Point 4. In particular the Authors should explain why these values are attributed to a specific geological setting such as a fore-arc basin and not with the strong metasomatic event that have affected the Kempirsai massif reported by several Authors (see for example Melcher et al. 1994, 1997).
Response 4: First, we do not claim that the geological setting was determined solely on the basis of oxygen fugacity data. We only state that the data we have received do not contradict the tectonic setting that was derived earlier (Melcher et al., 1997). We make relevant references in the text. This geodynamic conclusion is also consistent with the geological position of the massif at the front of the Magnitogorsk paleo island arc.
Secondly, with regard to the “major metasomatic event” that some authors write about, we do not agree with this point of view and think that the discussion on this topic goes far beyond the topic of our article. In several previous papers (in Minerals, Mineralium Deposita, Mineralogy and Petrology), we considered the issues of the genesis of podiform chromitites, discussed in detail various existing models, and substantiated our own point of view. Articles are available at the link (https://cloud.mail.ru/public/UzZP/ZMNr8Xaen)
In the same works, it was shown that the smallest newly formed grains of Cr-spinel inside olivine and enstatite are nucleated on structure defects of these minerals caused by plastic deformation, where they are often associated with tiny amphibole grains. At the same time, we do not deny the important role of the fluid components present in the mantle material. They could contribute to the weakening of peridotites, as well as the transfer of some elements to inclusions in microconcentrations.
